# Human exploration strategically balances approaching and avoiding uncertainty

Yaniv Abir[1]*, Michael Neil Shadlen[2,3], Daphna Shohamy[1,2]

[1]Department of Psychology, Columbia University, New York, United States; [2]Zuckerman Mind Brain Behavior Institute, and Kavli Institute for Brain Science, Columbia University, New York, United States; [3]Department of Neuroscience and Howard Hughes Medical Institute, Columbia University, New York, United States

## eLife Assessment

This study presents an **important** investigation of how people approach and avoid uncertainty, with a particular focus on the effects of overall uncertainty. They find that individuals approach uncertainty to a point, but when uncertainty is particularly high, they avoid it. The results are interpreted under a cognitive cost-resource rational framework. The methods are **convincing**, using appropriate and current methodologies.

*For correspondence:
y.abir@ucl.ac.uk

Competing interest: The authors declare that no competing interests exist.

## Abstract

A central purpose of exploration is to reduce goal-relevant uncertainty. Consequently, individuals often explore by focusing on areas of uncertainty in the environment. However, people sometimes adopt the opposite strategy, one of avoiding uncertainty. How are the conflicting tendencies to approach and avoid uncertainty reconciled in human exploration? We hypothesized that the balance between avoiding and approaching uncertainty can be understood by considering capacity constraints. Accordingly, people are expected to approach uncertainty in most cases, but to avoid it when overall uncertainty is highest. To test this, we developed a new task and used modeling to compare human choices to a range of plausible policies. The task required participants to learn the statistics of a simulated environment by active exploration. On each trial, participants chose to explore a better-known or lesser-known option. Participants generally chose to approach uncertainty; however, when overall uncertainty about the choice options was highest, they instead avoided uncertainty and chose to sample better-known objects. This strategy was associated with faster decisions and, despite reducing the rate of observed information, it did not impair learning. We suggest that balancing approaching and avoiding uncertainty reduces the cognitive costs of exploration in a resource-rational manner.

## Introduction

The purpose of exploration is to reduce uncertainty about the aspects of one's environment that are goal relevant or otherwise important. Yet, devising an optimal strategy to reduce uncertainty is known to be very difficult (*Cohen et al., 2007*; *Schulz and Gershman, 2019*; *Sutton and Barto, 2018*), especially for agents with limited memory and processing capacities. A heuristic strategy that is often efficient for exploration is focusing on the parts of the environment that one is most uncertain about. This principle of approaching uncertainty has been applied in a range of fields, including statistics (*MacKay, 1992*; *Sebastiani and Wynn, 2000*), artificial intelligence (*Badia et al., 2020*; *Bellemare et al., 2016*; *Pathak et al., 2017*; *Raposo et al., 2021*), and cognitive theories of human exploration (*Schulz and Gershman, 2019*; *Schwartenbeck et al., 2019*). Indeed, humans have been shown to approach uncertainty when learning about rewards in the environment through trial and error (*Schulz*

*and Gershman, 2019*; *Speekenbrink and Konstantinidis, 2015*; *Wilson et al., 2014*; *Wu et al., 2022*).

However, there are also many examples of uncertainty avoidance in the decision-making of humans and animals. Uncertainty avoidance has been documented in situations where resolving uncertainty may reveal negative outcomes and news (*Ahmadlou et al., 2021*; *Botta et al., 2020*; *Eilam and Golani, 1989*; *Glickman and Sroges, 1966*; *Gordon et al., 2014*; *Gigerenzer and Garcia-Retamero, 2017*; *Golman et al., 2017*), or may make overcoming a conflict in motivation more difficult (*Carrillo and Mariotti, 2000*; *Golman et al., 2017*). When the goal is to maximize immediate rewards, choosing the most rewarding option often entails avoiding more uncertain options (*Trudel et al., 2021*; *Wilson et al., 2014*).

How are the two conflicting tendencies to approach and avoid uncertainty reconciled when exploring? To answer this question, we must address gaps in the literature about exploration at two levels of analysis. At the computational level, it is unclear what might compel individuals to avoid uncertainty instead of approaching it, bar holding goals other than attaining knowledge. Indeed, avoiding uncertainty reduces the rate of information intake, and so might result in poorer learning. At the algorithmic level, we lack an understanding of how individuals compute uncertainty to make exploratory choices. Computing uncertainty exactly is complicated and often intractable. Several candidate algorithms for approximating the computation of uncertainty have been suggested (*Schulz and Gershman, 2019*), but evidence as to their use by humans is still preliminary.

It is the complexity of choosing based on uncertainty, set against the limited processing and memory capacities that are inherent to human cognition, that motivated our hypotheses regarding both the algorithmic and computational questions. First, we charted a hypothesis space of plausible algorithms for computing uncertainty and making exploratory choices (*Schulz and Gershman, 2019*), starting with the optimal but complex, and ending with simple approximations. Second, we hypothesized that the complexity of choosing what to explore, even when using approximate algorithms, is the key factor explaining why and when individuals might avoid uncertainty in exploration. Adhering to the goal of approaching uncertainty may well be an efficient policy for an agent with unlimited cognitive resources. Since humans have finite memory systems, inference bandwidth, and time, it stands to reason that they would try to conserve these resources by regulating their exploration (*Lieder and Griffiths, 2020*), possibly by selectively avoiding uncertainty. Following this insight, we examined exploratory choices as a function of two factors affecting the difficulty of making an exploratory choice: participants' overall uncertainty about choice options (*Schulz and Gershman, 2019*), and forgetting.

We developed a task in which participants made multiple exploratory choices, incrementally building knowledge toward a distant goal (*Figure 1*). Importantly, participants were given reward feedback only at the end of a round and not after every trial, allowing us to focus on choices made to accumulate knowledge, rather than choices driven by the need to exploit available rewards. Seeking ecological validity, we designed a task that posed a challenging exploration problem for participants, requiring that they infer and remember the values of multiple latent parameters from repeated experience (*Hartley, 2022*; *Lieder and Griffiths, 2020*). The task could nonetheless be captured by a few mathematical expressions, allowing for the derivation of the optimal exploration policy. This optimal policy served as a basis for a quantitative analysis of participants' choices and reaction times with the aim of identifying the algorithm driving their exploratory choices (*Anderson, 1990*; *Chater and Oaksford, 1999*; *Waskom et al., 2019*).

## Results

194 participants from a pre-registered (*Abir et al., 2021*) sample were recruited to complete up to 22 rounds of the exploration task over four online sessions. The task simulated a room with four tables, with two decks of cards on each table (*Figure 1a–b*). If a card was flipped, it was revealed to be, for example, either orange or blue (each round used a different pair of colors). The proportion of orange vs. blue cards, π, differed between the two decks on each table. Participants' goal was to learn $sgn(\pi_1 - \pi_2)$, or which deck had more orange (blue) cards on each table. We will denote this term, which serves as the learning desideratum for participants, as $\theta$.

The task begins with an exploration phase, followed by a test phase. On each trial of the exploration phase, participants chose which of two tables to explore and then revealed one card from a deck

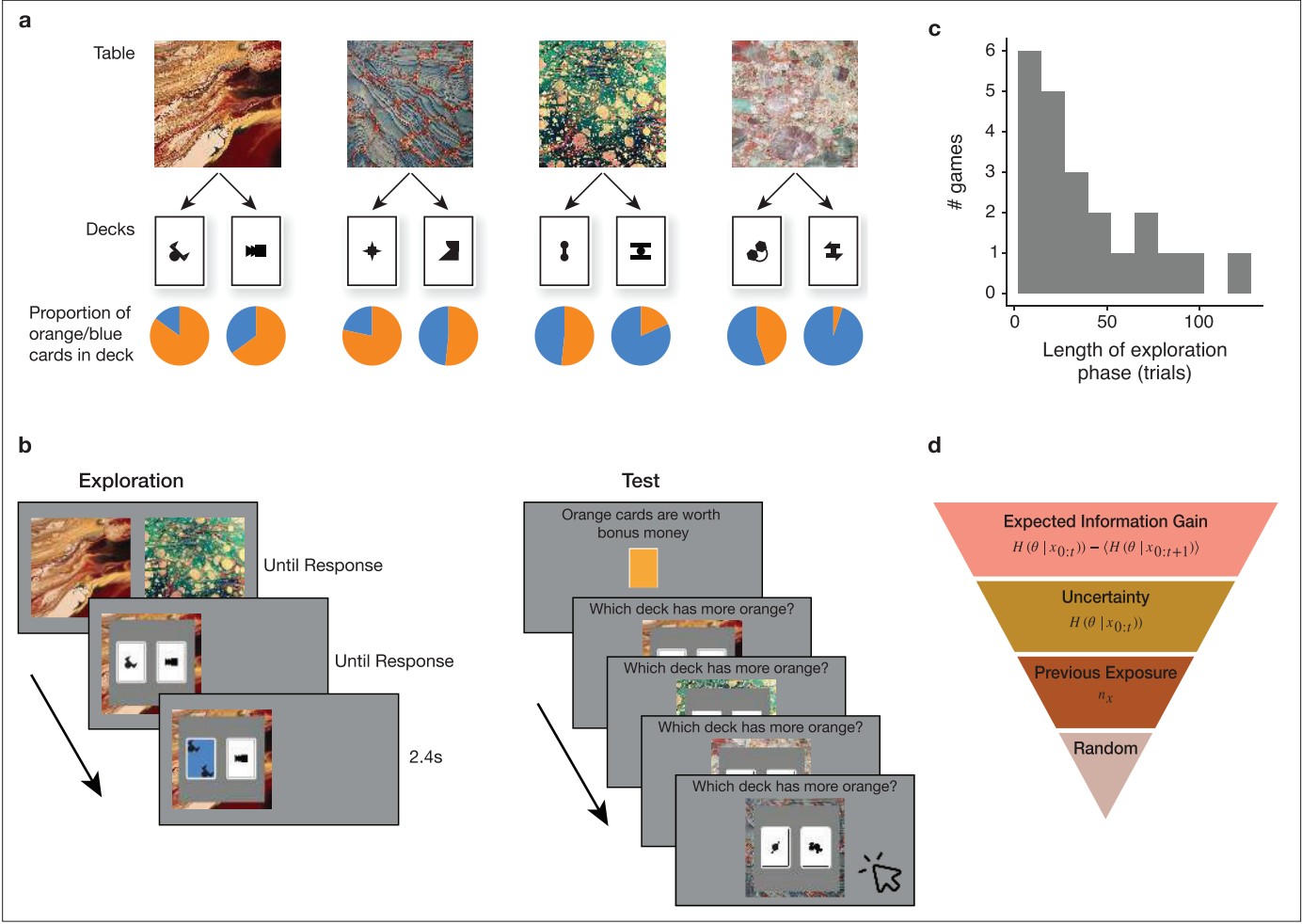

**Figure 1.** Examining exploration strategy in relation to uncertainty in an incremental learning task. (**a**) Structure of the task. Participants explored four tables, each containing two decks with different proportions of blue/orange cards. The goal was to learn the difference in proportions of the decks on each table. (**b**) The two phases of the task - exploration and test. On a single exploration trial (left), participants chose between two tables, and then sampled a card from one of the decks on that table, observing its color. After a random number of exploration trials, participants were tested on their knowledge (right). A color was designated as rewarding, and participants then chose the deck with the highest proportion of the rewarding color on each table. They were rewarded for correct test-phase choices and received no reward during exploration. (**c**) Histogram of round lengths. Participants played 22 rounds. The length of exploration in each round followed a shifted geometric distribution, such that the test was equally likely to occur following any trial after the first 10. (**d**) We considered a hierarchy of strategies for choosing which table to explore. The normatively prescribed strategy is to choose the table affording maximal expected information gain. This is the table for which the next card is expected to maximally decrease uncertainty (measured as entropy $H$) about the value of the goal-relevant latent parameter $\theta$, given observations thus far $x$. A simpler strategy is to choose the table with the maximum uncertainty, as it does not necessitate computing an expectation over the next observation. An even simpler heuristic is to equate previous exposure and choose the table with the least previous observations $n_x$. Even though these three strategies vary considerably in complexity, they are all uncertainty-approaching on average. Lastly, people may be random explorers.

on that table (*Figure 1b*). Participants were instructed that the exploration phase would be followed by a test phase after a random number of trials (drawn from a geometric distribution to discourage pre-planning, *Figure 1c*). They were further instructed that one of the colors would be designated as rewarding at the beginning of the test phase. During the test phase, participants were asked to indicate which deck had more of the rewarding color on each table (*Figure 1b*). They also rated their confidence in the choice. For every correct test-phase choice, they received $0.25. Crucially, they received no reward during exploration. Participants' only incentive during the exploration phase was to maximize their confidence about the value of $\theta$.

## Three hypothetical strategies derived by rational analysis

To explain how participants chose between tables in the exploration phase, we first asked how an optimal agent might solve the problem of choosing which table to explore on each trial of the task. We limited our consideration to strategies that optimize learning only for the next trial, since a globally optimal strategy is intractable for this task (*Schulz and Gershman, 2019*; *Sutton and Barto, 2018*). We started by deriving the optimal strategy and progressively simplified it to generate two additional strategies. While they differ in the level of complexity they assume, all three strategies direct an agent using them to approach the option they are more uncertain about.

The optimal strategy, given by the expression at the top of *Figure 1d*, is choosing the table affording maximal expected information gain (EIG; *Gureckis and Markant, 2012*; *MacKay, 1992*; *Yang et al., 2016*). EIG is the difference between the uncertainty in the value of the learning desideratum, $\theta$, given observed cards $x_{0:t}$, and the expected uncertainty after observing the next card on trial $t + 1$. In other words, EIG is the amount of uncertainty resolvable on the next trial.

Computing the second term in the EIG expression requires averaging over future unseen outcomes, which may be beyond the ability of participants. As an alternative, they might avoid computing this term by simply choosing the table they were more uncertain about at the moment of making the choice (*Figure 1d*, second tier; *Schulz and Gershman, 2019*). While this strategy has intuitive appeal, computing uncertainties may still be too complicated for human participants. An even simpler heuristic is given on the third tier of *Figure 1d*: choosing the table with the least prior exposure (*Auer, 2002*; *Schulz and Gershman, 2019*), measured as the number of already observed cards $n_x$. Since on average additional observations result in lower uncertainty, this strategy is an approximate way to approach the more uncertain table. Finally, participants might explore at random, rather than in a directed manner (*Daw et al., 2006*; *Schulz and Gershman, 2019*; *Wilson et al., 2014*).

## Test phase performance validates observation model

To relate the three hypothesized strategies to participants' behavior, we assumed a model of participants' beliefs about the goal-relevant parameter $\theta$ and the mechanism by which they updated these beliefs. We used a Bayesian observer model which forms beliefs about $\theta$ based on the actual card sequence each participant observed and updates these beliefs according to Bayes' rule (*Figure 2*). On its own, the Bayesian observer does not predict participants' exploration choices, but only models the process of inference from observation.

Before evaluating the hypothesized exploration strategies, we sought to validate the assumptions of the Bayesian observer model. To this end, we related the predictions of the Bayesian observer model to participants' choices during the test phase. We predicted that test accuracy should be greater when the Bayesian observer model had low uncertainty about $\theta$ at the end of the learning phase. The data supported this prediction (*Figure 3*). Using a multilevel logistic regression model, we confirmed that test accuracy was strongly related to the Bayesian observer's uncertainty b=−5.59, 95% posterior interval (PI)=[-6.25,–4.95] (all effect sizes given in original units, full model and coefficients reported in *Appendix 3—table 1*). Participants' reports of confidence after making a correct choice also followed the Bayesian observer's uncertainty b=−4.04, 95% PI=[-4.50,–3.56]. After committing errors, participants' reported confidence was lower overall b=−1.09, 95% PI=[-1.27,–0.92], and considerably less dependent on Bayesian observer uncertainty, interaction b=−3.10, 95% PI=[-3.76,–2.46] (*Figure 3b*, *Appendix 3—table 2*).

## Uncertainty is the best predictor of exploratory choice

To evaluate the three exploration strategies, we tested whether participants' exploration-phase choices could be predicted from the difference between the two tables that were presented as choice options in each of the hypothesized quantities. We fit the data with a multilevel logistic regression model for each strategy (*Appendix 3—tables 3–5*). In a formal comparison of the three models, we found that uncertainty was the best predictor of exploratory choices, as indicated by a reliably better prediction metric (*Figure 4*). Accordingly, the difference in uncertainty for the table presented on the right versus the table presented on the left (Δ-uncertainty) predicts participants' choices. Δ-EIG provides a poorer fit to choices, and Δ-exposure is anti-correlated with choice, in contradiction of the exposure hypothesis. We confirmed that our analysis approach can recover the true model generating a simulated dataset (*Figure 4—figure supplement 1*). Furthermore, simulations showed that

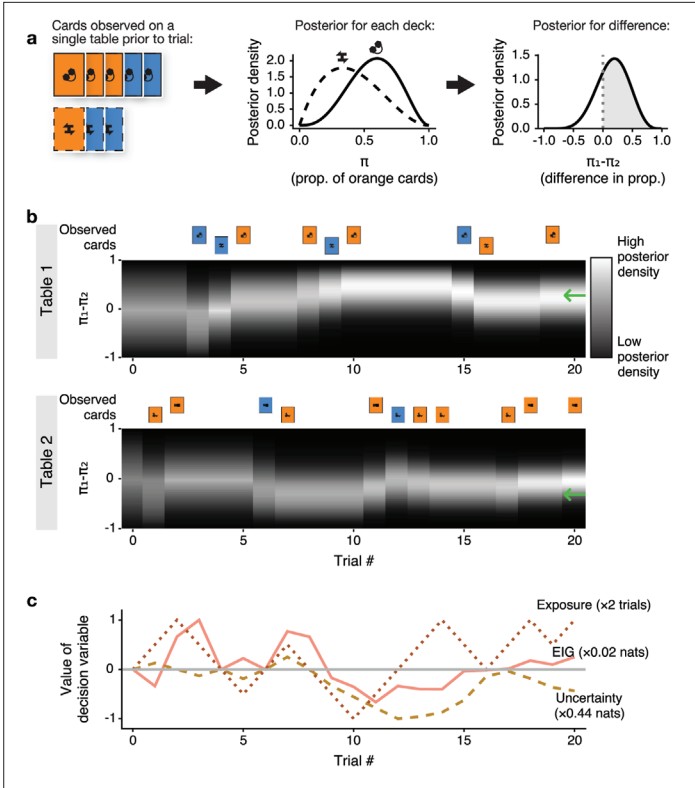

**Figure 2.** Hypothetical strategies make differing predictions for exploratory choice behavior. We computed the three quantities hypothesized to drive exploratory choices using a Bayesian observer model. To illustrate this process, we plot the derivation of Bayesian belief on a single trial (**a**) and across multiple trials (**b, c**). For visualization, we use a simplified version with two tables only. **a** depicts the Bayesian observer's belief about a single table on a single trial. Given a sequence of previously observed cards (left), the Bayesian observer forms posterior beliefs about the proportion of orange cards in each deck (center). These beliefs are expressed as Beta distributions. From these, it is possible to derive a belief about the difference in the proportion of orange cards between the two decks $\pi_1 - \pi_2$ (right). The probability that $\pi_1 > \pi_2$ is given by the proportional size of the area marked in gray (0.74 in this example). (**b**) Depicts the same process over a series of 20 trials. The observed card sequence for each table is presented at the top of each panel. The matching belief state about $\pi_1 - \pi_2$ is plotted below it as an evolving posterior density in white (high) and black (low). The green arrows mark the true value of $\pi_1 - \pi_2$ for that round. As the round progresses, belief converges towards the true value and becomes more certain. (**c**) The three choice strategies prescribe different table choices on most trials. The difference between table 1 and table 2 in each of the three quantities (EIG, uncertainty, and exposure) is plotted for each trial. This difference is the hypothesized decision variable for choosing between tables 1 and 2. A positive value indicates a preference for exploring table 1, and a negative value indicates a preference for table 2. The three variables are normalized to facilitate visual comparison.

The online version of this article includes the following figure supplement(s) for figure 2:

**Figure supplement 1.** Correlations between the three strategies.

uncertainty is a sufficient predictor of choice. Simulated datasets generated by uncertainty-driven agents recreated the entire set of qualitative and quantitative results (*Figure 4—figure supplement 2*). The simulations demonstrate that the surprising negative correlation between choice and Δ-exposure is an epiphenomenon of uncertainty-driven exploration: agents repeatedly return to harder-to-learn tables, gaining more exposure to them precisely because they remain more uncertain about these tables.

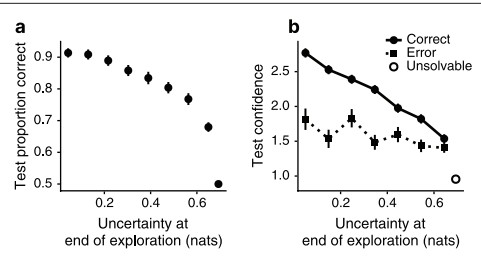

**Figure 3.** The Bayesian observer model is validated by participants' accuracy and confidence on the test phase. (**a**) Participants were accurate when an exploration phase ended with low uncertainty and performed at chance level when the phase ended with high uncertainty. (**b**) Participants' confidence on correct choices fell with rising uncertainty. Confidence on error trials did not depend as much on Bayesian observer uncertainty. When a test question was unsolvable because no evidence was observed on each deck during exploration, participants had very low confidence. Data presented as mean values ± 1 SE, n=194 participants.

The online version of this article includes the following figure supplement(s) for figure 3:

**Figure supplement 1.** Matching results in the preliminary sample.

## Participants systematically change their exploration strategy according to overall uncertainty

We next asked whether participants' strategy of exploring by approaching uncertainty is modulated by the state of their knowledge when making an exploratory choice. Specifically, we examined how participants' overall uncertainty about the two options they could choose to explore on a given trial changed the way they explored (*Figure 5a*). Since table choice options were presented at random, participants sometimes had to choose between tables they already knew a lot about, and sometimes between tables they were very uncertain about. When overall uncertainty was high, the choice between tables had to be made with very little evidence. Note that from a normative perspective, choice should follow the difference in uncertainty between options and should not be influenced by overall uncertainty.

We found a systematic deviation in exploration strategy in relation to overall uncertainty. When overall uncertainty for the two choice options was below a certain threshold, participants chose the more uncertain table, as expected. However, when overall uncertainty was above the threshold, they chose the less uncertain table, thereby slowing the rate of information intake (*Figure 5b and c*).

We validated this observation using a multilevel piecewise-regression model, allowing for the influence of Δ-uncertainty on choice to differ below and above a fitted threshold of overall uncertainty. We observed a positive relationship between Δ-uncertainty and choice below the threshold b=0.97, 95% PI=[0.83,1.11], but above the threshold, we found that the influence of Δ-uncertainty on choice became strongly negative (interaction b=−4.3e+02, 95% PI=[−5.4e+02,−3.4e+02]). The group-average

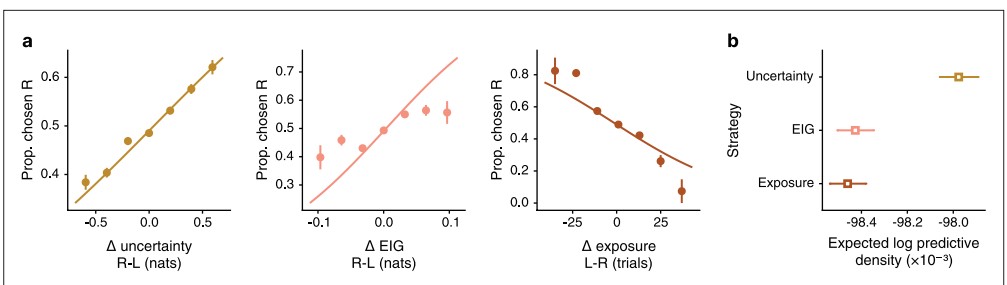

**Figure 4.** Uncertainty is the best predictor of choice. (**a**) On each plot, the difference in the hypothesized quantity between the two tables presented on each trial is plotted against actual choices of the table presented on the right. For each plot, the relevant hypothesis predicts a positive smooth curve. Δ-uncertainty, plotted on the left, matches this prediction better than Δ-EIG (center). The relationship between Δ-exposure (right) and choice is negative, rather than the hypothesized positive correlation. (**b**) Quantitative model comparison confirms this observation. Out of the three hypothesized strategies, uncertainty has the highest approximate expected log predictive density (using PSIS LOO; see Methods). Data presented as mean values ± 1 SE, n=194 participants.

The online version of this article includes the following figure supplement(s) for figure 4:

**Figure supplement 1.** Fitting simulated data successfully recovers the underlying strategy.

**Figure supplement 2.** Uncertainty is a sufficient predictor of choice.

**Figure supplement 3.** Matching results in the preliminary sample.

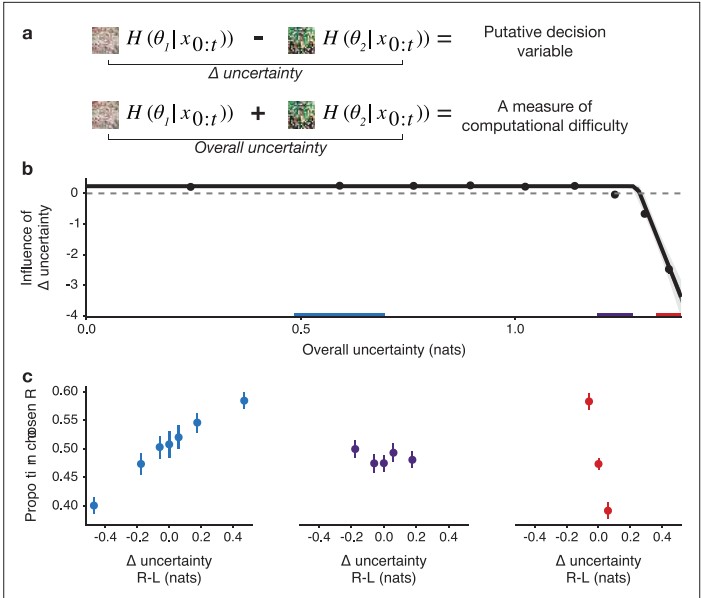

**Figure 5.** Participants approach vs. avoid Δ-uncertainty as a function of overall uncertainty. (**a**) While the Δ-uncertainty is the decision variable identified above, overall uncertainty, defined as the sum of uncertainty for both tables, is a measure of decision difficulty. (**b**) The influence of Δ-uncertainty on choice differed markedly below and above a threshold of overall uncertainty. Below a certain threshold of overall uncertainty, estimated as a free parameter, Δ-uncertainty had a significant positive effect on choice. Above this threshold of overall uncertainty, the influence of Δ-uncertainty became strongly negative. Points denote mean posterior estimate from regression models fitted to binned data, error bars mark 50% PI. The solid line depicts the prediction from a piecewise regression model capturing the non-linear relationship and estimating the threshold, with darker ribbon marking 50% PI and light ribbon marking 95% PI. Data from three regions of overall uncertainty marked in color are plotted in (**c**) For low overall uncertainty (blue), participants tend to choose the table they are more uncertain about, as normatively prescribed. But that relationship is broken for medium levels of overall uncertainty (purple). For high overall uncertainty (red), participants strongly prefer to choose the table they are less uncertain about, thereby slowing down the rate of information intake. Data plotted as mean ± SE, n=194 participants.

The online version of this article includes the following figure supplement(s) for figure 5:

**Figure supplement 1.** Matching results in the preliminary sample.

**Figure supplement 2.** No correlation between overall uncertainty and Δ-uncertainty.

threshold was estimated to be 1.28 nats of overall uncertainty (95% PI=[1.27, 1.29]; *Appendix 3— table 6*), leaving 21.58% of trials in the high overall uncertainty range (95% PI=[20.12,24.45]). This bias in exploration cannot be viewed merely as a noisier version of optimal performance. Rather, it constitutes a systematic modulation of exploration strategy on about a fifth of the trials.

## Costs and benefits of strategically avoiding uncertainty

What motivates participants to systematically avoid learning about more uncertain objects? By the standards of an ideal agent, uncertainty avoidance is clearly suboptimal, as it reduces the rate of observed information, and thus the potential capacity to learn. We hypothesized that the limited processing and memory capacities that are inherent to human cognition and set it apart from the optimal agent are the reason for uncertainty avoidance. To test this hypothesis, we conduct a cost-benefit analysis of uncertainty avoidance in the following sections. We ask whether uncertainty avoidance is associated with costs to learning, and whether it affords any benefits in managing cognitive effort.

### Tendencies to approach vs. avoid uncertainty are associated with test performance

Since efficient learning is the purpose of exploration, we asked how the tendencies to approach uncertainty and avoid it when overall uncertainty is high affect learning as reflected in performance

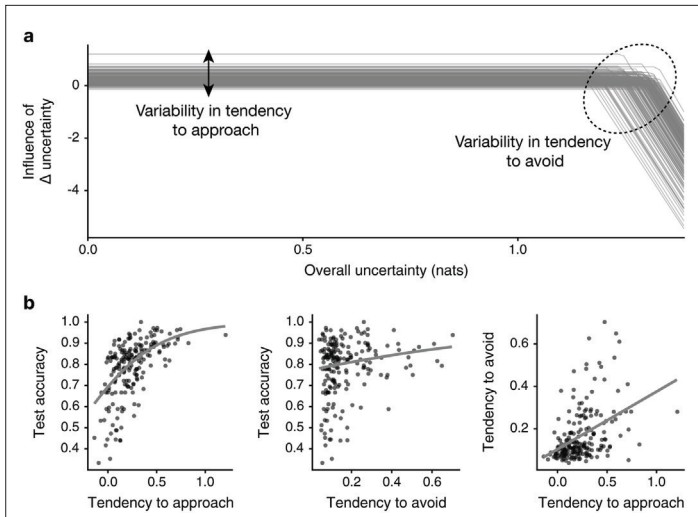

**Figure 6.** Learners benefit from approaching uncertainty, but are not penalized for avoiding it. (**a**) We observe substantial individual differences in strategy. Replotting *Figure 5e* separately for each participant highlights variation in the baseline tendency to approach uncertainty, as well as in the degree of avoidance when overall uncertainty is high. (**b**) A stronger baseline tendency to approach uncertainty (left) predicts better test performance, such that participants unable to approach uncertainty also perform poorly. Test performance shows a weak positive correlation with avoidance when overall uncertainty is high (middle), since learners who approach uncertainty also tend to avoid it under high uncertainty (right). Uncertainty avoidance is quantified as the triangular area above the piecewise regression line in panel **a**.

The online version of this article includes the following figure supplement(s) for figure 6:

**Figure supplement 1.** Matching results in the preliminary sample.

**Figure supplement 2.** Individual differences in the use of Δ-EIG and Δ-exposure: pre-registered sample.

**Figure supplement 3.** Individual differences in the use of Δ-EIG and Δ-exposure: preliminary sample.

at test. If approaching uncertainty is the only rational exploration policy, then participants who tend to approach uncertainty to a greater degree should learn more and perform better at the test, while participants with a strong tendency to avoid uncertainty should learn less and perform worse at the test, since they are choosing to forgo valuable information as they explore.

To test these predictions, we examined individual differences in exploration strategy in relation to test performance. We found that participants' baseline tendency to approach uncertainty predicted better performance at test b=2.96, 95% PI=[2.67,3.25] (*Figure 6b*; *Appendix 3—table 7*).

In contrast, the relationship between the tendency to avoid uncertainty and test performance was more nuanced. In both samples, participants who were more inclined to approach uncertainty also tended to avoid it when overall uncertainty was high ($r=0.43$, $p=5.42\times10^{-10}$). Accordingly, avoidance was positively correlated with test performance at the population level b=1.18, 95% PI=[0.80, 1.58] (*Figure 6b*; *Appendix 3—table 8*; see Methods for parameter estimation). However, once we adjusted for the tendency to approach, avoidance was reliably associated with worse test performance b=−0.83, 95% PI=[-1.28,–0.40] (*Appendix 3—table 9*).

Taken together, these findings suggest that avoidance, on its own, hinders learning. Yet in our samples, it was the better learners who also engaged in avoidance, implying that avoiding uncertainty when overall uncertainty is high may serve a complementary role that benefits those who already learn efficiently.

## Strategic exploration involves costly deliberation

To understand the costs involved in exploration, we asked whether making exploratory choices in this task involves prolonged deliberation. If that is the case, and exploratory choices are guided by Δ-uncertainty, we reasoned that decisions should require longer deliberation when the absolute value of Δ-uncertainty is small (*Palmer et al., 2005*; *Shushruth et al., 2022*). To test this prediction, we fit the data with a generative model of choice and RTs. We used a sequential sampling model,

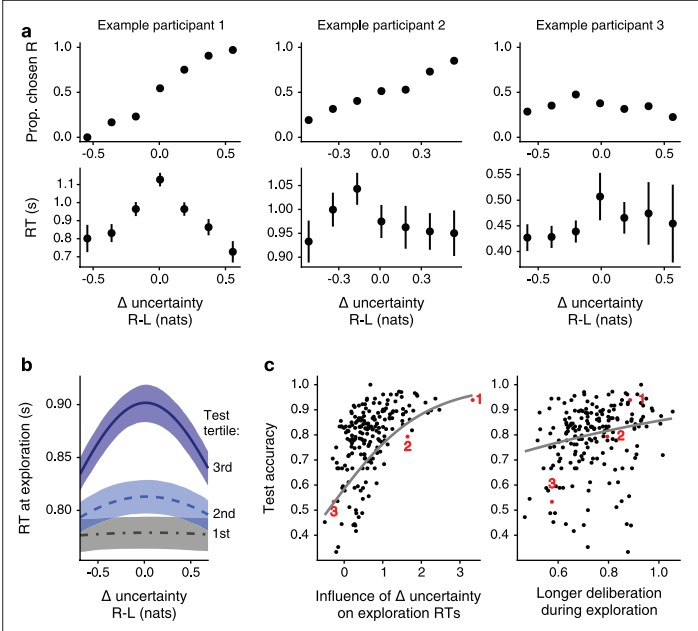

**Figure 7.** Individuals who spend time deliberating during exploration make strategic choices and learn well. Participants varied not only in the pattern of their choices, but also in their RTs. (**a**) Data from three example participants. The relationship of choice and RTs with Δ-uncertainty weakens from left to right. Data plotted as mean ± SE. (**b**) These individual differences were captured by a sequential sampling model, explaining choices and RTs as the interaction between participants' efficacy of deliberating about Δ-uncertainty and their tendency to deliberate longer vs. make quick responses. Plotting model predictions, we observe a U-shaped dependence of RTs on Δ-uncertainty for participants whose performance at test was in the top accuracy tertile. This characteristic u-shape is indicative of decisions made by prolonged deliberation. This relationship is weaker for participants in the bottom two test accuracy tertiles. Such participants also exhibit shorter RTs overall. Lines mark mean predictions from a sequential sampling model fit by tertiles for visualization, ribbons denote 50% PIs. (**c**) Correlating the sequential sampling model parameters with test performance confirms these observations. Participants with a stronger dependence of RT on Δ-uncertainty perform better at test, as do participants who deliberate longer for the sake of accuracy. Example participants from **a** are marked in red. Lines are mean predictions from a logistic regression model.

The online version of this article includes the following figure supplement(s) for figure 7:

**Figure supplement 1.** Matching results in the preliminary sample.

which explains decisions as the outcome of a process of sequential sampling that stops when the accumulation of evidence satisfies a bound. This model explains RTs as jointly influenced by participants' efficacy in deliberating about Δ-uncertainty, and their tendency to deliberate longer vs. make quick responses (*Ratcliff and McKoon, 2008*; *Shadlen and Kiani, 2013*; *Shadlen and Shohamy, 2016*). One prediction of sequential sampling theory is that greater deliberation efficacy should be manifested as greater dependence of RT on absolute Δ-uncertainty (*Palmer et al., 2005*).

We found that RTs indeed varied in relation to the absolute value of Δ-uncertainty as expected b=0.69, 95% PI=[0.58,0.78] (*Appendix 3—table 10*). Crucially, a stronger dependence of RT on the absolute value of Δ-uncertainty predicted better performance at test (drift-rate and test performance association b=0.81, 95% PI=[0.58,1.07]). We further found that participants who tended to deliberate longer for the sake of accuracy also tended to perform better at test (bound height and test performance association b=1.46, 95% PI=[0.58,2.34]; *Figure 7c*, *Appendix 3—table 11*). In summary, participants who were better at deliberating about uncertainty during exploration and who deliberated for longer performed better at test. Thus, making good exploratory choices that lead to efficient learning involves prolonged deliberation.

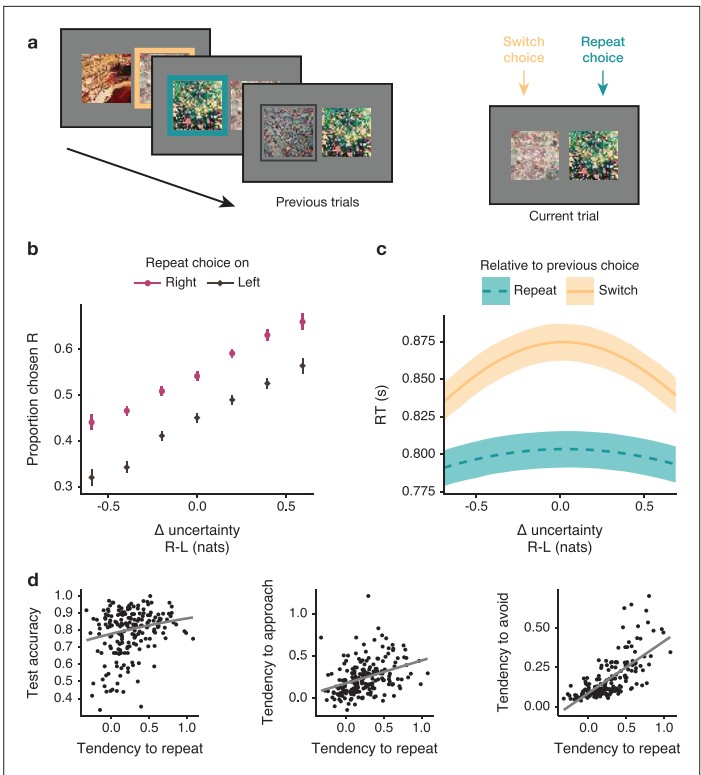

**Figure 8.** Participants tend to repeat previous choices instead of deliberating over uncertainty. (**a**) On a given trial, one table has been chosen more recently than the other (frames denote previous choices). In the example, the green table had been chosen more recently; hence, it is designated the repeat option and the other table the switch option. (**b**) Participants tend to choose the table displayed on the right more often when it is the repeat option than when it is the switch option. Data plotted as mean ± SE, n=194 participants. (**c**) When choosing a repeat option, participants' RTs are shorter and less dependent on Δ-uncertainty. Lines mark mean predictions from a sequential sampling model, ribbons denote 50% PIs. (**d**) Participants who tended to repeat their previous choice also tended to perform better at test (left), were more likely to have a stronger baseline tendency to approach uncertainty (middle), and a stronger tendency to avoid uncertainty when overall uncertainty is high (right). Regression lines are plotted for visualization.

The online version of this article includes the following figure supplement(s) for figure 8:

**Figure supplement 1.** Matching results in the preliminary sample.

## Deliberation is reduced by choice repetition

One reason participants may avoid uncertainty—rather than approach it—when overall uncertainty is high is that doing so reduces the need for prolonged deliberation. Unfortunately, we could not test for such a benefit by directly comparing RTs as a function of overall uncertainty, as overall uncertainty is related to the difficulty of making an exploratory choice. With a single independent variable, deconfounding the effect of difficulty from the strategies used to ameliorate it is impossible. Fortunately, we could take advantage of a conceptually related but independent tendency we observed in our dataset to examine the benefits of reduced deliberation times.

As in many learning tasks, participants in our task tended to repeat their previous choice (*Wu et al., 2022*), a tendency that was independent of Δ-uncertainty or overall uncertainty. We observed that participants generally preferred to re-choose the table they had last chosen (*Figure 8b*). We corroborated this with a multilevel regression model controlling for the effects of Δ-uncertainty and overall uncertainty b=0.50, 95% PI=[0.42,0.59] (*Appendix 3—tables 12 and 13*). Crucially, the tendency to repeat choices was also reflected in RTs, which for repeat choices were less related to Δ-uncertainty (b=−0.32, 95% PI=[-0.43,–0.22]). We also found that participants tended to make repeat choices more quickly rather than deliberate longer (b=−0.05, 95% PI=[-0.05,–0.04]; *Figure 8c*, *Appendix 3—table 14*).

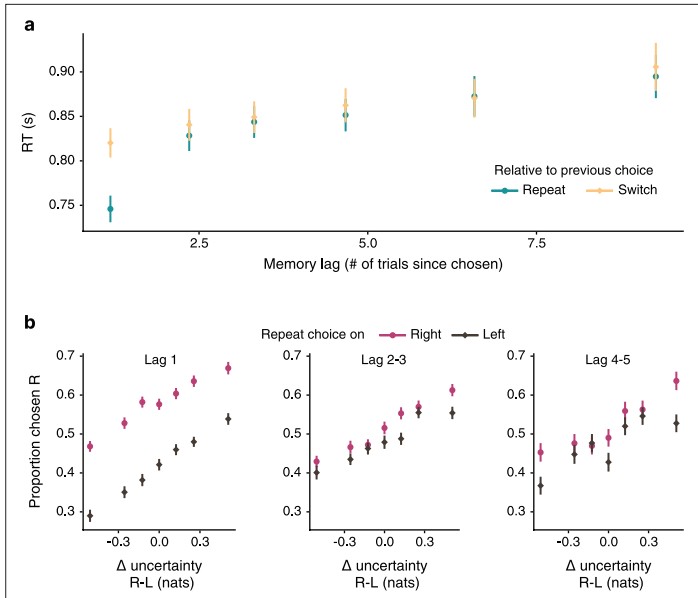

**Figure 9.** Forgetting is associated with random choice rather than a systematic bias. (**a**) Memory lag, defined as trials since last choice, serves as a proxy for forgetting and contributes to the difficulty of making an exploratory choice. RTs rise with memory lag. The RT advantage for repeat choices disappears with higher memory lag. (**b**) With higher memory lag, choices become less dependent on Δ-uncertainty, as indicated by flatter curves. The tendency to repeat the last choice is also diminished with memory lag. Both effects amount to choice becoming more random due to forgetting. Data plotted as mean ± SE, n=194 participants.

The online version of this article includes the following figure supplement(s) for figure 9:

**Figure supplement 1.** Matching results in the preliminary sample.

As in other aspects of exploration strategy, we observed considerable individual differences in the tendency to repeat previous choices. These differences were associated with the uncertainty-based aspects of exploration discussed above (*Figure 8d*). Participants with a general tendency to repeat choices show stronger uncertainty avoidance when overall uncertainty is high, indicating that these two conceptually related strategies also co-occur in the population $r=-0.60$, 95% PI=[-0.74,–0.43] (*Appendix 3—table 12*). Furthermore, the tendency to repeat previous choices is associated with better test performance, logistic regression b=0.09, 95% PI=[0.07,0.11] (*Appendix 3—table 15*). The tendency to repeat is also correlated with a stronger baseline tendency to approach uncertainty $r=0.32$, 95% PI=[0.17,0.46] (*Appendix 3—table 12*), which was shown above to be correlated with test performance. Thus, while from a normative point of view, repeating the previous choice appears to be a context-insensitive heuristic, in practice, participants who use this strategy do not learn any worse.

## Forgetting as a conceptual control

Explaining participants' deviation from the optimal exploration strategy as rational is interesting only to the extent that rationality is not a foregone conclusion. Is the alternative hypothesis of a failure in decision making also a priori plausible? We turned to forgetting as a second source of difficulty in our task and a conceptual control condition. Due to the random presentation of choice options, there was variability in the number of trials passed since either of the presented tables was last explored. We assumed that choosing between tables that had not been explored for a long time is more difficult than between tables for which evidence has been recently observed. Indeed, we found that RTs were longer with a larger lag, indicating greater difficulty of making a choice (log normal regression b=0.02, 95% PI=[0.02, 0.03]; *Figure 9a*, *Appendix 3—table 16*). Furthermore, we observed that exploration choices on trials with a greater lag depended less on Δ-uncertainty b=−0.08, 95% PI=[-0.11,–0.04], and that the tendency to repeat the last chosen table on these trials was also diminished b=−0.13, 95% PI=[-0.15,–0.11] (*Figure 9b*, *Appendix 3—table 17*). Finally, on trials with a large lag,

the difference in RTs between making a repeat and a switch choice disappeared, interaction b=0.02, 95% PI=[0.02,0.03] (*Figure 9a*, *Appendix 3—table 16*). These patterns suggest that prior evidence is forgotten with increasing lag and that as a consequence, exploration becomes more random. Hence, in contrast to the systematic effect of overall uncertainty, forgetting results in a failure to make principled exploratory choices.

## Discussion

We examined the cognitive computations behind exploratory choices using a paradigm that encourages incremental learning in the service of a distant goal. We found that uncertainty played an important role in guiding participants' choices about how to sample their environment for learning. In general, participants chose to learn more about the options they were more uncertain about. However, when overall uncertainty was especially high, participants instead avoided the more uncertain options and sampled the options they already knew more about. In addition, we found that participants tended to repeat previous choices. Together, this pattern suggests that participants systematically balance approaching and avoiding uncertainty while exploring.

Examining individual differences in exploration and learning revealed both costs and benefits of avoiding uncertainty. We found that strategically avoiding uncertainty was associated with a detriment to learning only when adjusting for the baseline tendency to approach uncertainty. We also found an association between the length of deliberation and learning efficiency. Participants who deliberated longer also learned better, and deliberation time could be shortened by repeating previous choices. Based on these results, we conclude that balancing approaching and avoiding uncertainty is a way to manage cognitive resources by regulating deliberation costs. In this sense, our results serve as an example of how human cognition is adapted to the inherent constraints of the human mind, consistent with the resource rationality framework (*Lieder and Griffiths, 2020*).

While the literature on exploration is expansive, the paradigm presented here extends it in important ways. Researchers of reinforcement learning have previously examined exploration in the context of reward-seeking decisions. Using paradigms such as an n-armed bandit task (*Schulz and Gershman, 2019*), it was demonstrated that humans don't always choose options they believe will yield the most reward, but also make random and directed choices with the aim of exploring other uncertain options (*Schulz and Gershman, 2019*; *Wilson et al., 2014*). Recently, studies using the bandit task have lent empirical support to the notion that exploration is difficult, as participants explore less under time pressure or cognitive load (*Brown et al., 2022*; *Otto et al., 2014*; *Cogliati Dezza et al., 2019*; *Wu et al., 2022*). Crucially, this literature has focused on cases where reward can be gained on each trial (*Brown et al., 2022*; *Cohen et al., 2007*; *Daw et al., 2006*; *Schulz and Gershman, 2019*; *Song et al., 2019*; *Tversky and Edwards, 1966*; *Wilson et al., 2014*; *Wu et al., 2022*). In such tasks, the motivation to exploit existing knowledge tends to dominate, making exploratory behavior rare and difficult to measure (*Findling et al., 2019*). In contrast, our task was designed to eliminate the immediate incentive to exploit current knowledge, allowing us to observe a large number of exploratory choices. With this increased experimental power, we were able to compare different algorithms approximating the goal of approaching uncertainty and describe how and when humans avoid uncertainty instead of approaching it.

Several previous studies of exploration inspired us to design a task with separate exploration and test phases. Using the 'observe or bet' paradigm, *Tversky and Edwards, 1966* examined how participants trade off exploration and exploitation on a trial-by-trial basis. A similar paradigm has been used to study when participants choose to end their exploration and how this decision affects learning (*Wulff et al., 2018*). The paradigm presented here extends these approaches, as it is crafted to reveal the strategy driving each exploration choice.

Exploration has also been studied in the information search literature (*Gureckis and Markant, 2012*; *Markant and Gureckis, 2014*; *Oaksford and Chater, 1994*; *Petitet et al., 2021*; *Rothe et al., 2018*; *Ruggeri et al., 2017*), which inspired our analysis approach. In most studies of this field, participants make decisions without relying on their memory, as the entire history of learning is displayed to them on screen (cf. related work in active sensing; *Yang et al., 2016*). This differs from our task, which places heavy demands on memory. Rather than treating capacity limitations as a source of noise and a nuisance to measurement, we find that the rational use of limited resources is central for successful exploration.

We observed considerable individual differences in exploration strategy, as would be expected in a complex task requiring memory-based learning and inference. In the face of such variability, one might question the prudence of drawing conclusions at the population level, since averages could obscure a range of idiosyncratic strategies. However, the strong correlation between individual differences in exploration and test performance mitigates this concern. It suggests that participants who were engaged with the task and able to learn from observation tended to adopt a strategy that systematically balanced approaching and avoiding uncertainty. The relationship between test performance and reaction times lends further mechanistic support to this idea. Notably, these individual differences also imply that this balance is shaped not only by the structure of the learning problem but also by characteristics of the learner. Future research is needed to identify and understand the cognitive or dispositional factors that underlie these individual differences.

Our theoretical analysis and experiments leave several open questions. One concerns the relationship between overall uncertainty and time on task: in our paradigm, overall uncertainty was correlated with the number of cards observed. Although our findings remain robust when trial number is included as a covariate in the regression models (see *Appendix 3—table 18*), future work could more directly disentangle these factors by orthogonalizing overall uncertainty and elapsed time. This might be achieved, for instance, by manipulating overall uncertainty within a game—such as by introducing new tables or altering outcome probabilities mid-round.

Another open question is the nature of the limitation driving participants to avoid uncertainty when overall uncertainty is high. This could be due to limitations in committing prior experiences to memory, inferring latent parameters from disparate experiences, retrieving prior knowledge, or estimating the uncertainty of existent knowledge. While the idea that decisions based on high overall uncertainty are more difficult has been raised previously (*Schulz and Gershman, 2019*; *Shafir, 1994*), an explanation grounded in cognitive mechanisms is still needed. Accordingly, the mechanism by which uncertainty avoidance ameliorates choice difficulty remains unknown.

One intriguing explanation for the source of difficulty and the way it is managed lies in the distinction between strategies dependent on remembering single experiences and those dependent on the incremental accumulation of knowledge in the form of summary statistics (*Collins et al., 2017*; *Collins and Frank, 2012*; *Daw et al., 2005*; *Knowlton et al., 1996*; *Plonsky et al., 2015*; *Poldrack et al., 2001*). Both strategies could contribute to performance in tasks such as ours. A participant may be encoding prior observations as single instances or summarizing them into a central tendency with a margin of uncertainty around it. Crucially, each strategy is associated with a different profile of cognitive resource use. Keeping track of individual experiences is much costlier than tracking a single expectation and a confidence interval around it (*Daw et al., 2005*; *Nicholas et al., 2022*) and more likely to incur costs when switching between exploring different tables. Prior work suggests individuals use single experiences or summary statistics according to the reliability of each strategy and the cost of using it (*Daw et al., 2005*; *Nicholas et al., 2022*). In our case, summary statistics may be perceived as unreliable when overall uncertainty is high, compelling participants to rely on committing individual experiences to working memory (*Bavard et al., 2021*; *Collins et al., 2017*; *Daw et al., 2005*; *Duncan et al., 2019*; *Poldrack et al., 2001*). Furthermore, recent work examining how humans make a series of dependent decisions demonstrates that the tension between remembering single experiences and discarding them in favor of summary statistics is accompanied by a tendency to revisit previous choices instead of switching to new alternatives (*Zylberberg, 2021*).

The questions we addressed here were partly motivated by the well-established observation that humans and animals often avoid uncertainty in various situations. Two broad categories of explanation for such avoidance have been proposed (*Golman et al., 2017*). First, individuals avoid resolving uncertainty when it could lead to negative news, for example by avoiding ambiguous prospects when making economic choices (*Ellsberg, 1961*; *Fox and Tversky, 1995*). An extension of this idea is dread avoidance (*Gigerenzer and Garcia-Retamero, 2017*; *Golman et al., 2017*). One might avoid resolving the uncertainty about a medical diagnosis to avoid the unpleasant affective response to the news, even if the information could be very useful in determining treatment. Relatedly, humans might avoid uncertainty as a by-product of pursuing a goal other than exploration. More uncertain options are often avoided for the sake of choosing immediately rewarding options (*Trudel et al., 2021*; *Wilson et al., 2014*) — why try an unknown dish, when your absolute favorite is on the menu? Lastly, uncertainty avoidance may be a strategy for managing conflict between different motivations,

or different mechanisms of action selection (*Carrillo and Mariotti, 2000*; *Golman et al., 2017*). For example, to maintain their diet, an individual might choose to avoid resolving the uncertainty about what snacks can be found in the office kitchen. Our findings highlight a different kind of strategic uncertainty avoidance. In our tasks, there were no negative consequences to learning about the color proportions of card decks, and no conflicting motivations. Rather, we explain participants' tendency to avoid uncertainty in terms of managing their limited cognitive resources.

The idea of a balance between approaching and avoiding uncertainty has conceptual parallels in other literatures. A group of relevant findings concerns how animals explore their proximal environment. A classic finding in rats is that when placed in a novel open arena, they alternate between the exploration strategy of walking around the arena (uncertainty approaching) and a strategy of returning to their initial position and pausing there (termed 'home base' behavior, which is uncertainty avoiding; *Eilam and Golani, 1989*). Relatedly, by using computational models to understand how rats use their whiskers to explore near objects, researchers have identified an alternation between uncertainty approaching and avoiding strategies (*Gordon et al., 2014*). Recent work in mice and primates has uncovered neural circuits driving exploration by framing the problem of exploration as striking a balance between approach and avoidance (*Ahmadlou et al., 2021*; *Botta et al., 2020*; *Ogasawara et al., 2022*). Our findings highlight the shared computational principles between human exploration in symbolic space and animal exploration of the physical environment and suggest that mechanisms involved in avoidance responses may also play a part in knowledge acquisition.

Finally, planning (*Hunt et al., 2021*), learning (*Gureckis and Markant, 2012*), and sensing (*Gordon et al., 2014*; *Yang et al., 2016*) are increasingly studied as active processes, situated within our environment and interacting with it. Understanding the complicated dynamics between agent and environment has been greatly facilitated by comparing behavior against the computational ideal of maximizing the amount of information observed (*Oaksford and Chater, 1994*; *Schulz and Gershman, 2019*; *Schwartenbeck et al., 2019*). The findings we present here suggest a modification to this computational premise. Rather than trying to uncover as much information as possible, the goal of human exploration may be to maximize the amount of information retained in memory, by modulating the rate and order of observed information.

## Methods
### Data collection and participants

A sample of 298 participants was recruited via Amazon MTurk to participate in four sessions of the exploration task. They were paid $3.60 per session, plus a performance-based bonus ($4.50 for the first session, $6 for later sessions). Additionally, a $2 bonus was paid out for completion of the fourth session. Participants were asked to complete the four sessions over the course of a week and were invited by email to each session after the first, as long as the data from their last session was not excluded according to the criteria we had specified (see below). All participants provided informed consent; all protocols were approved by the Columbia University Institutional Review Board (#AAAI1488).

The first session was terminated early for 89 participants due to recorded interactions with other applications during the experiment or failure to comply with instructions. An additional 32 sessions played by participants who had successfully completed the first session were excluded for the same reasons. One participant was excluded after reporting technical problems with stimulus presentation in the second session. Twenty-seven further sessions were excluded for failure to sample cards from both decks, a prerequisite for learning on which participants were instructed as part of the training. Altogether, data from 194 participants was included in the analyzed sample (120 female, 72 male, 2 other gender, average age 29.63, range 20–48). This sample included 194 first sessions, 156 second sessions, 129 third sessions, and 116 fourth sessions.

Before running this experiment, we pre-registered (*Abir et al., 2021*) a sample size of 190 participants satisfying our exclusion criteria. We chose this number to be three times larger than a preliminary sample of N=62 participants, which provided the dataset we used to develop our analysis approach and pipeline, and first identify exploration strategies as described above. Results for the preliminary sample are provided in figure supplements.

## Task design and procedure

On each round of the exploration task, participants were presented with a simple environment of four tables with two decks of cards on each table. Tables were distinguished by unique colorful patterns and decks by geometric symbols that did not repeat within an experimental session. The hidden side of each card was painted in one of two colors, with a unique color pair for each round. The proportion of colors in each deck was determined pseudo-randomly (see Appendix 1), resulting in variability in the difference in proportion between each deck pair - the learning desideratum of this task.

At the beginning of each round, participants were first presented with the color pair for the round, and then with the table-deck assignments. Participants then had to pass a multiple-choice test on the table-deck assignment, making sure they remembered the structure of the task before proceeding to explore. Failing to get a perfect score on this test resulted in repeating this phase. The exploration phase then commenced. Trial structure for the exploration phase is depicted in *Figure 1b*. The lengths of the exploration phases varied from round to round. They were sampled from a geometric distribution with rate $\frac{1}{44}$, shifted by 10 trials. The same list of round lengths was used for all participants, but their order was randomized.

Following the exploration phase, participants were tested on their learning. They were presented with the rewarding color for this round, and then had to indicate which deck had a greater proportion of that color on each table (*Figure 1c*). After answering this question for each of the four tables, they rated their confidence in each of the four choices on a 1–5 Likert scale. Participants were then told whether each of the test choices was correct, and the true color proportions for the two decks on each table were presented to them as 10 open cards.

The first session started with extensive instructions explaining the structure of each of the two phases of the task and clearly stating the learning goal. Participants were also instructed on the independence of color proportion within each deck pair, necessitating sampling from both decks to succeed in the task. The instructions also included training on how to make the relevant choices in each of the two stages. A quiz followed the instruction phase, and participants had to repeat reading the instructions if they had given the wrong response to any question on this quiz.

Each session started with a short practice round (12–19 trials). Data from this round was excluded from analysis. In the first session, participants then played three more rounds and in later sessions, five more rounds, for a total of 18 experimental rounds.

## Data analysis

Analysis was performed using Julia 1.4.2. Hierarchical regression models were fitted using the Stan probabilistic programming language 2.30.1 (*Carpenter et al., 2017*), using the interface supplied by the brms package version 2.16.1 (*Bürkner, 2017*), running on top of R 4.1.2. The complete computing environment was packaged as a Docker image, which can be used to reproduce the entire analysis pipeline. Sequential sampling models were fitted on a separate Docker image (*Chuan-Peng et al., 2022*) containing HDDM 0.8 (*Wiecki et al., 2013*) running on top of python 3.8.8.

### Bayesian observer

Each of the three hypothesized strategies for exploration postulates a different summary statistic of prior learning as the driver of exploratory choice. To derive these summary statistics, we first had to construct a model of prior learning. We chose a simple Bayesian observer model (*Behrens et al., 2007*; *Yang et al., 2016*). Like our participants, this model's goal was to learn $\theta = sgn(\pi_1 - \pi_2)$ from observed outcomes $x_{0:t}$. It did so by placing a probabilistic prior over the value of each $\pi_i$, updating it after every observation according to Bayes' rule, and solving for $\theta$ using the rules of probability. The result is a posterior distribution capturing the agent's expectation of the value of $\theta$, and their uncertainty about the expectation. This process is depicted in *Figure 2* for two tables and their matching pairs of decks.

This computation can be put into formulaic form as follows. At the beginning of a round, the Bayesian observer places a flat Beta distribution prior on the proportion of colors in each of the eight decks:

$$\pi_i \sim Beta(1, 1)$$

After observing a card, this prior would be updated to form a posterior distribution. Since the posterior of a Beta prior and a Bernoulli observation likelihood is also a Beta distribution, the posterior has a simple analytic form: after completing t trials, observing $c_i$ cards of one color and $t - c_i$ cards of the other color, the posterior would be:

$$\pi_i | x_{0:t} \sim Beta(1 + c_i, 1 + t - c_i)$$

We can then find the probability that $\theta = 1$, that is that $\pi_1 > \pi_2$, by calculating the probability that $\pi_2$ is smaller than a given $\pi_1 = z$, and integrating over $z$, the possible values of $\pi_1$:

$$P(\theta = 1 | x_{0:t}) = \int_0^1 f_{\pi_1 | x_{0:t}}(z) F_{\pi_2 | x_{0:t}}(z) dz$$

where $f$ is the Beta probability density function, $F$ is the Beta cumulative density function, and $x_{0:t}$ are observations thus far. We computed the value of this integral numerically using the Julia package QuadGK.jl. Finally, $\theta$ can only take two values, and so

$$P(\theta = -1 | x_{0:t}) = 1 - P(\theta = 1 | x_{0:t})$$

## Computing hypothesized decision variables

The theory of decision making defines a decision variable as the quantity evaluated by the decision maker in order to choose between two choice options (**Shadlen and Kiani, 2013**). The difficulty of the decision should scale with the absolute value of the decision variable. Each of the three hypothesized strategies is defined by a specific summary statistic of prior learning that might serve as the decision variable for an exploratory choice. The three summary statistics are given in **Figure 1e**.

Both EIG and uncertainty are derived from the uncertainty of the posterior for $\theta$ as defined above. We quantified uncertainty as the entropy of the posterior belief (**MacKay, 1992**; **Oaksford and Chater, 1994**; **Yang et al., 2016**):

$$H(\theta | x_{0:t}) = - \sum_{\theta = -1, 1} P(\theta | x_{0:t}) ln P(\theta | x_{0:t})$$

Entropy takes the unit of nats, ranging from 0 should the participant be absolutely sure about the value of $\theta$ for both table choice options, to 0.69 when they know nothing about a table. This is the equivalent of 1 bit of information, were we to replace the natural logarithm with a base 2 logarithm.

Because entropy values in this paradigm — as in any finite information learning problem — are bounded, overall uncertainty and Δ-uncertainty are not independent. Specifically, when overall uncertainty is very high or very low, the possible range of Δ-uncertainty is necessarily constrained. Importantly, however, their values are not correlated (**Figure 5—figure supplement 2**).

## Model combining Δ-uncertainty and overall uncertainty

To quantify how the influence of Δ-uncertainty on choice varied with overall uncertainty, we fit a multilevel piecewise logistic regression model. This model estimated a threshold in overall uncertainty, treated as a free parameter, and allowed the slope of Δ-uncertainty on choice to differ below and above this threshold. Below the threshold, a positive slope reflects a tendency to approach uncertainty; above the threshold, a negative interaction captures the tendency to avoid Δ-uncertainty with higher values of overall uncertainty.

To examine how individual differences in exploration strategy relate to test performance, we extracted participant-level parameters from the multilevel models. Each participant's tendency to approach uncertainty was directly estimated in the piecewise regression model described above. To capture the tendency to avoid uncertainty when overall uncertainty was high, we computed, for each individual, the area above the declining segment of their regression line presented in **Figure 6a**. This metric encompasses both individual differences in the threshold parameter and the interaction term.

## Estimating multilevel Bayesian models for inference

The regression coefficients and PIs reported here were all estimated using multilevel regression models accounting for individual differences in behavior. We used regularizing priors for all coefficients of

**Table 1.** Regularizing priors used in regression models.

| Type of coefficient | Prior for logistic and ordered-logistic regression | Prior for lognormal regression (RTs; following Schad et al., 2019) |
| --- | --- | --- |
| Intercept | normal(0,1) (not applicable for ordered logistic models; *Bürkner and Vuorre, 2019*) | normal(–0.25, 0.5) |
| Group-level effects of predictors | normal(0,1) | normal(0, 0.5) |
| Scale of by-participant terms | normal(0,1) | normal(0, 0.01) |
| Correlation matrices for by-participant terms | LKJ(2) | LKJ(2) |

Prior distributions are given in Stan syntax. All predictors used in models were centered and scaled prior to fitting, so that the same priors can apply to all parameters.

interest to facilitate robust estimation (*Table 1*). For RT data, we selected informative priors for the intercept term in the regression (capturing the grand average of RTs) following established recommendations (*Schad et al., 2021*). For predicting choices, we used logistic regression, for confidence ratings we used ordinal-logistic regression, and for average RTs we used log-normal regression. We estimated these models with Hamiltonian Monte Carlo implemented in the Stan probabilistic programming language using the R package brms. Three Monte Carlo chains were run for each model, collecting 1000 samples each after a warm-up period of at least 1000 samples (warm-up was extended if convergence had not been reached). Sequential sampling models were estimated using slice sampling, implemented in the python package HDDM. Four Monte Carlo chains were run for each model, collecting 2000 samples each after a warm-up period of at least 2000 samples. Convergence for both model types was assessed using the $\hat{R}$ metric, and visual inspection of trace plots. R syntax formulae and coefficients for covariates for all models mentioned in the main text are reported in Appendix 3.

## Sequential sampling model of reaction times

To draw inference from participants' RTs we turned to the sequential sampling theory of deliberation and choice. This theory encompasses a family of models in which decisions arise through a process of sequential sampling that stops when the accumulation of evidence satisfies a threshold or bound (*Palmer et al., 2005*; *Shadlen and Kiani, 2013*). From this family of models, we chose to use the drift diffusion model (DDM) to fit our data, as it is very well described and extensively studied (*Ratcliff and McKoon, 2008*; *Shadlen and Kiani, 2013*). The DDM explains RTs as the culmination of three interpretable terms. The first is the efficacy of a participant's thought process in furnishing relevant evidence for the decision - in our case, the efficacy of choosing according to Δ-uncertainty (the drift rate in DDM parlance). The second term governs the participant's speed-accuracy tradeoff by determining how much evidence they require to commit to a decision. This can also be thought of as how long a participant is willing to deliberate when a decision is difficult (bound height). Finally, the portion of the RT not linked to the deliberation process is captured by a third term (non-decision time). Since behavior was considerably different when overall uncertainty was high, DDM models were fit excluding trials with overall uncertainty above the participant's estimated threshold.

## Model evaluation

We compared the models of choice and RTs to alternative models, either reduced or expanded (see Appendix 3). We used the LOO R package to perform approximate leave-one-out cross-validation for models implemented in Stan. This method uses Pareto-smoothed importance sampling to approximate cross-validation in an efficient manner (*Vehtari et al., 2017*). Models implemented in HDDM were compared using the DIC metric. We also performed recovery analysis and posterior predictive checks for our models, making sure they capture the theoretically important qualitative features of the data.

## Acknowledgements

We thank the Shohamy lab, Christopher A Baldassano, and Gabriel M Stine for their insightful discussion of the project. We are thankful for the support of the Stan user community, especially Matti Vuorre and Paul Bürkner. We are grateful for funding support from the NSF (award #1822619 to DS), NIMH/NIH (#MH121093 to DS) and the Templeton Foundation (#60844 to DS).

## Additional information

### Funding

| Funder | Grant reference number | Author |
| --- | --- | --- |
| Division of Information and Intelligent Systems | 1822619 | Daphna Shohamy |
| National Institute of Mental Health | MH121093 | Daphna Shohamy |
| Templeton World Charity Foundation | 60844 | Daphna Shohamy |

The funders had no role in study design, data collection and interpretation, or the decision to submit the work for publication.

### Author contributions

Yaniv Abir, Conceptualization, Formal analysis, Investigation, Writing – original draft; Michael Neil Shadlen, Daphna Shohamy, Conceptualization, Supervision, Writing – review and editing

### Author ORCIDs

Yaniv Abir ⬤ https://orcid.org/0000-0002-3725-9011
Michael Neil Shadlen ⬤ https://orcid.org/0000-0002-2002-2210

### Ethics

All participants provided informed consent; all protocols were approved by the Columbia University Institutional Review Board, protocol #AAAI1488.

Reviewer #1 (Public review): https://doi.org/10.7554/eLife.94231.3.sa1
Author response https://doi.org/10.7554/eLife.94231.3.sa2

## Additional files

### Supplementary files

MDAR checklist

### Data availability

All data, analysis scripts, and computational environment are available at https://osf.io/6zyev.

The following dataset was generated:

| Author(s) | Year | Dataset title | Dataset URL | Database and Identifier |
| --- | --- | --- | --- | --- |
| Abir Y, Shadlen MN, Shohamy D | 2022 | Human Exploration Strategically Balances Approaching and Avoiding Uncertainty | https://osf.io/6zyev | Open Science Framework, 6zyev |

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

# Appendix 1

## Full details of task and procedure

### Recruitment

Participants were recruited from the pool of Amazon Mechanical Turk (MTurk) vetted by https://cloudresearch.com (*Hauser et al., 2023*; *Litman et al., 2017*). We further restricted enrollment to participants with an approval rating higher than 95% and at least 100 prior jobs completed. Enrollment was also restricted to participants registered as being 18–35 years old. Participants were presented with an ad for a multi-session study, describing base pay and the performance-dependent bonus. The ad stated that we were only looking for participants who were willing to complete all four sessions, and that the task required undivided attention.

After accepting the task, participants were directed to a website running the experiment, which was coded using jsPsych 6.0.4 (*de Leeuw, 2015*).

### Instructions and training

At the beginning of the first session, participants were thoroughly instructed about the task and practiced each of its phases. First, the two-phase structure and the learning goal were introduced. Participants then were shown an example table with two decks on it. They practiced choosing a deck on a single table. Participants were instructed that only the symbol on the deck determined its identity, while its location (which was randomly determined) did not matter. Participants then completed ten practice trials limited to choosing decks on a single table, with the goal of figuring out the difference in proportions of colors between the decks. In this practice, one deck had 7/10 cards of the rewarding color, and the other 9/10 cards. After practicing choosing decks, participants were told which had more of the rewarding color, and it was explained that the differences could be more difficult to figure out in the game itself. Next, participants were told that while both decks may have a majority of cards of the same color. As such, they would have to sample from both decks to learn which had more of each color. This point was demonstrated by presenting ten cards from two decks - and asking (i) which deck had a majority of color 1 cards (both did), (ii) which had a majority of color 2 cards (none did), (iii) which deck had more of color 1 than the other (one of the decks did), and (iv) which deck had more of color 2 than the other (the other deck did). A failure to give a correct answer on all four questions prompted a repetition of this section.

Participants were then introduced to choosing a table before making a deck choice. As practice, they were tasked with revealing a single card from each deck on two tables over four trials. Failing to do so resulted in participants having to repeat this section.

Next, the test phase was introduced. Participants were instructed that a particular deck had more of the rewarding color in it. They then had to choose that deck on the following test screen. Having successfully done so, they received the same visual feedback they would receive in the actual task - 10 cards of each deck were presented to them, demonstrating the true proportion of colors in each deck. A message was displayed alongside this demonstration, stating the accuracy of their choice.

Before starting work on the main task, participants were reminded that the test phase could commence on any given trial. They then answered six multiple-choice questions about the structure of the task and their goal. If they got any of these wrong, they were instructed on the correct answer and then had to repeat the quiz. After successfully completing the quiz, they played the whole task over a 20-trials-long practice round. For this practice, only two tables were included in the set of stimuli. Finally, after completing this practice round, they continued to play four more rounds of the full task with four tables.

Each of the following three sessions began with a short reminder of the instructions. Participants then completed five rounds of the task. The first round was always a short one (12–19 trials, these are the four lowest numbers drawn from the geometric distribution of round lengths), and was treated as a practice round in analysis, that is data from this round was discarded.

## Procedure for single round

### Familiarization
At the beginning of each round, participants were presented with each of the tables included in the round, and the two decks associated with each table. They were then tested on these associations: they were shown a table, and two decks next to it, and had to indicate which of the two belonged to the table. Failure to answer correctly on all eight trials resulted in a repetition of the introduction. Following the introductions of tables and decks, participants were shown the two card colors used in this round. They were reminded about their goal, and then commenced to play the learning phase.

### Learning phase
Each trial of the learning phase began with a 500 ms period during which a fixation cross was displayed at the center of the screen. Two tables were then presented as choice options. Choice options were chosen for each trial at random, and the left-right presentation of the two choice options was also determined randomly. Participants indicated their choice of table by pressing either the 'd' or 'k' keys. Next, the unchosen table was removed from the screen, and a frame with the same pattern as the chosen table was presented for a period of 1000 ms. This was followed by a 500 ms presentation of a fixation cross at the center of this frame. Then, the two decks associated with the table appeared in the frame (deck location was determined randomly). Using the same keys, participants chose to reveal a card on one of the decks. A short animation was played showing the deck being shuffled, and a single card was flipped to reveal its color. The colored card remained on screen for 2400 ms. A 1700 ms inter-trial interval followed each trial.

### Test phase
At the end of the learning phase, participants were instructed that they would now be tested on their learning and were shown the color designated as the rewarding color for this round. They then chose the deck they believed had more of the rewarding color on each table. After making all four choices, participants were shown their choice on each table and asked to rate their confidence that their choice was indeed the correct one.

### Memory test
Following the confidence ratings, participants were tested on their memory of table and deck associations. They were shown each deck participating in the round and had to indicate which of the four tables it belonged to. Participants received no feedback for this memory test.

### Feedback
Next, participants received feedback for their test-phase choices. For each table, they were shown ten cards drawn from each deck. The cards represented the true proportion of colors in the deck. They were reminded which deck they had chosen and were told whether that was the correct choice. After observing this for each of the four tables, participants were told how much bonus money they earned in this round.

## Debriefing
At the end of the experiment, after collecting demographic information, participants were asked about any strategy they may have implemented to remember the card colors better, to choose between decks and between tables in the learning phase, and between decks in the test phase. Lastly, they were asked if anything in the instructions remained unclear. These responses were evaluated for any use of external aids, such as pen and paper, and for any technical difficulties.

## Compliance and attention checks
We implemented several measures to incentivize participants to devote their undivided attention to the task. First, the task ran in full screen mode, and if participants chose to exit it, a warning message was shown explaining that this study only runs in fullscreen mode. Additionally, refreshing the webpage and right-clicking on it were disabled. Instances where the participant interacted with

another application on their computer were recorded by jsPsych, and when this occurred, a warning message was displayed on screen.

Participants had to make each learning-phase choice within 3000 ms. Failing to do so resulted in the display of a warning message asking them to choose more quickly. Additionally, if a reaction time (RT) of less than 250 ms was recorded on three consecutive choices, a warning message asking participants to comply with instructions was displayed on screen.

When more than 10 warning messages of any kind had been displayed, the session terminated, and participants were asked to return the job to MTurk. Participants were paid for terminated sessions but were not invited to following sessions.

# Appendix 2

## Preliminary sample

### Data collection and participants

A preliminary sample of 70 participants was recruited via MTurk to participate in four sessions of the task. Task design was identical to that later used for the pre-registered sample. Recruitment was not limited to the https://cloudresearch.com approved sample, since data collection predated the proliferation of fake worker accounts on MTurk (*Hauser et al., 2023*).

The first session was terminated early for seven participants due to recorded interactions with other applications during the experiment, or failure to comply with instructions. An additional eight sessions played by participants who had successfully completed the first session were excluded for the same reasons. Five further sessions were excluded for failure to sample cards from both decks, a prerequisite for learning on which participants were instructed as part of the training. Altogether, data from 62 participants was included in the analyzed sample (33 female, 28 male, 1 other gender, average age 29.10, range 20–38). This sample included 62 first sessions, 45 second sessions, 33 third sessions, and 29 fourth sessions.

### Results

Results from the pre-registered sample largely replicate the results we first observed in the preliminary sample (see matching figure supplement for each figure). Two points of divergence are observed. In the preliminary sample, we didn't see a significant correlation between the tendency to avoid uncertainty when overall uncertainty is high and test performance, while in the larger pre-registered sample, we observed a positive significant correlation. This difference could be due to sample size, as the correlation is weak; we should not overinterpret it.

A second point of divergence regards the plotting of predicted RTs by test performance tertile. While both in the preliminary sample and the pre-registered sample, we find that the bound height estimated by the DDM is predictive of test performance (*Appendix 3—table 14*), in the preliminary sample, this relationship does not translate to a monotonic rising of bound height when comparing participants by test performance tertile (*Figure 8—figure supplement 1*). We hesitate to interpret any non-linear relationship between the bound height and test performance, given the small size of the sample, the divergence from the pre-registered sample, and the complexity of the models involved.

# Appendix 3

**Appendix 3—table 1.** Test accuracy as a function of the final uncertainty in the exploration phase.

| | Pre-registered sample | | Preliminary sample | | |
|---|---|---|---|---|---|
| | (12.379 trials, 194 participants) | | (3482 trials, 62 participants) | | |
| Term | Median | 95% PI | Median | 95% PI | Units |
| Predictors | | | | | |
| Intercept | 0.86 | [0.61, 1.11] | 0.90 | [0.47, 1.32] | logit |
| Final uncertainty | -5.59 | [-6.25,-4.95] | -5.69 | [-6.86,-4.68] | logit / nats |
| Participant-wise variability | | | | | |
| SD of intercept | 1.58 | [1.38, 1.82] | 1.45 | [1.13, 1.89] | |
| SD of final uncertainty | 7.49 | [6.54, 8.61] | 6.83 | [5.33, 8.88] | |
| Correlation of intercept and final uncertainty | -0.68 | [-0.84,-0.46] | -0.81 | [-0.97,-0.40] | |

The model can be summarized with the following R syntax formula:
$$accuracy \sim 0.5 + 0.5 * inv\_logit(1 + final > uncertainty + (1 + final > uncertainty|participant)),\ \text{where}$$

**inv_logit** is the inverse logit function. This functional form limits predicted accuracy between 0.5 and 1.0, since guessing-level accuracy on a two-alternative forced-choice test is 0.5. Since accuracy is a binary variable, this regression was fit with a Bernoulli likelihood.

**Appendix 3—table 2.** Test confidence as a function of the final uncertainty in the exploration phase.

| | Pre-registered sample | | Preliminary sample | | |
|---|---|---|---|---|---|
| | (12,007 trials, 194 participants) | | (3362 trials, 62 participants) | | |
| Term | Median | 95% PI | Median | 95% PI | Units |
| Predictors | | | | | |
| Threshold 1 | -2.22 | [-2.44,-2.00] | -2.11 | [-2.49,-1.72] | |
| Threshold 2 | -0.44 | [-0.67,-0.23] | -0.20 | [-0.58, 0.17] | |
| Threshold 3 | 1.31 | [1.08, 1.53] | 1.60 | [1.22, 1.98] | |
| Threshold 4 | 3.11 | [2.88, 3.34] | 3.58 | [3.18, 3.98] | |
| Final uncertainty | -2.48 | [-2.93,-2.06] | -2.32 | [-3.11,-1.56] | logit / nats |
| Choice accuracy | 1.09 | [0.92, 1.27] | 1.05 | [0.71, 1.39] | logit |
| Final uncertainty × choice accuracy | -3.10 | [-3.76,-2.46] | -3.20 | [-4.60,-1.93] | logit / nats |
| Participant-wise variability | | | | | |
| SD of intercept | 1.45 | [1.31, 1.63] | 1.43 | [1.19, 1.73] | |
| SD of final uncertainty | 2.10 | [1.73, 2.52] | 2.12 | [1.39, 2.97] | |
| SD of choice accuracy | 0.81 | [0.64, 0.99] | 0.84 | [0.52, 1.25] | |
| SD of uncertainty × accuracy | 2.11 | [1.46, 2.82] | 3.02 | [1.58, 4.59] | |
| Correlation of intercept and uncertainty | 0.23 | [0.03, 0.41] | 0.07 | [-0.26, 0.39] | |
| Correlation of intercept and accuracy | -0.20 | [-0.39, 0.01] | 0.02 | [-0.33, 0.40] | |
| Correlation of uncertainty and accuracy | -0.57 | [-0.81,-0.30] | -0.51 | [-0.83,-0.03] | |
| Correlation of intercept and uncertainty × accuracy | 0.27 | [-0.01, 0.52] | 0.28 | [-0.11, 0.62] | |

*Appendix 3—table 2 Continued on next page*

*Appendix 3—table 2 Continued*

|  | Pre-registered sample | | Preliminary sample | |
|---|---|---|---|---|
| Correlation of uncertainty and uncertainty × accuracy | 0.68 | [0.34, 0.91] | 0.26 | [–0.24, 0.70] |
| Correlation of accuracy and uncertainty × accuracy | -0.84 | [-0.95,–0.61] | -0.23 | [–0.66, 0.35] |

The model can be summarized with the following R syntax formula:
*confidence ∼ 0 + final uncertainty ∗ accuracy + (1 + final uncertainty ∗ accuracy|PID)*. This model was fit as an ordered-logistic regression, with four threshold variables since confidence was rated on a 5-point Likert scale (*Bürkner and Vuorre, 2019*).

**Appendix 3—table 3.** Exploration-phase choices as a function of Δ-uncertainty.

|  | Pre-registered sample | | Preliminary sample | | |
|---|---|---|---|---|---|
|  | (146,766 trials, 194 participants) | | (41,009 trials, 62 participants) | | |
| Term | Median | 95% PI | Median | 95% PI | Units |
| Predictors |  |  |  |  |  |
| Intercept | -0.04 | [–0.10, 0.02] | -0.03 | [–0.11, 0.06] | logit |
| Δ-uncertainty | 0.89 | [0.76, 1.03] | 1.01 | [0.70, 1.30] | logit / nat |
| Participant-wise variability |  |  |  |  |  |
| SD of intercept | 0.39 | [0.35, 0.43] | 0.36 | [0.30, 0.45] |  |
| SD of Δ-uncertainty | 0.89 | [0.79, 1.00] | 1.11 | [0.91, 1.38] |  |
| Correlation of intercept and Δ-uncertainty | -0.05 | [–0.20, 0.12] | 0.07 | [–0.21, 0.31] |  |

The model can be summarized with the following R syntax formula: *table on right chosen ∼ 1 + Δuncertainty +*

*(1 + Δ uncertainty|participant)*. This model was fit as a logistic regression.

**Appendix 3—table 4.** Exploration-phase choices as a function of Δ-EIG.

|  | Pre-registered sample | | Preliminary sample | | |
|---|---|---|---|---|---|
|  | (146,766 trials, 194 participants) | | (41,009 trials, 62 participants) | | |
| Term | Median | 95% PI | Median | 95% PI | Units |
| Predictors |  |  |  |  |  |
| Intercept | -0.04 | [–0.09, 0.02] | -0.03 | [–0.12, 0.07] | logit |
| Δ-EIG | 10.12 | [8.00, 12.47] | 12.50 | [7.97, 16.91] | logit / nat |
| Participant-wise variability |  |  |  |  |  |
| SD of intercept | 0.39 | [0.35, 0.43] | 0.35 | [0.29, 0.43] |  |
| SD of Δ-EIG | 1.21 | [1.07, 1.37] | 1.48 | [1.21, 1.82] |  |
| Correlation of intercept and Δ-EIG | -0.02 | [–0.17, 0.14] | -0.11 | [–0.35, 0.16] |  |

The model can be summarized with the following R syntax formula: *table on right chosen ∼ 1 + ΔEIG +*

*(1 + Δ EIG|participant)*. This model was fit as a logistic regression.

**Appendix 3—table 5.** Exploration-phase choices as a function of Δ-exposure.

|  | Pre-registered sample | Preliminary sample |
|---|---|---|
|  | (146,766 trials, 194 participants) | (41,009 trials, 62 participants) |

*Appendix 3—table 5 Continued on next page*

*Appendix 3—table 5 Continued*

| Term | Pre-registered sample | | Preliminary sample | | |
|------|---------|---------|---------|---------|-------|
| | Median | 95% PI | Median | 95% PI | Units |
| Predictors | | | | | |
| Intercept | -0.04 | [–0.10, 0.02] | -0.03 | [–0.13, 0.07] | logit |
| Δ-exposure | -0.03 | [-0.04,–0.02] | -0.03 | [-0.05,–0.02] | logit / trial |
| Participant-wise variability | | | | | |
| SD of intercept | 0.39 | [0.35, 0.43] | 0.36 | [0.29, 0.44] | |
| SD of Δ-exposure | 0.05 | [0.04, 0.06] | 0.05 | [0.04, 0.06] | |
| Correlation of intercept and Δ-exposure | 0.16 | [–0.01, 0.32] | -0.08 | [–0.36, 0.21] | |

The model can be summarized with the following R syntax formula: ***table on right chosen ∼ 1 + Δ exposure +***

***(1 + Δ exposure|participant)***. This model was fit as a logistic regression.

**Appendix 3—table 6.** Exploration-phase choices as a function of Δ-uncertainty and overall uncertainty.

| Term | Pre-registered sample (146766 trials, 194 participants) | | Preliminary sample (41,009 trials, 62 participants) | | |
|------|---------|---------|---------|---------|-------|
| | Median | 95% PI | Median | 95% PI | Units |
| Predictors | | | | | |
| Intercept | -0.04 | [–0.10, 0.02] | -0.03 | [–0.13, 0.07] | logit |
| Δ-uncertainty | 0.97 | [0.83, 1.11] | 1.12 | [0.83, 1.42] | logit / nat |
| Δ-uncertainty × overall uncertainty | -428.44 | [-536.60,–339.27] | -444.27 | [-559.73,–353.23] | logit / nat² |
| Transformed threshold α | 2.52 | [2.40, 2.64] | 2.33 | [2.17, 2.49] | a.u. |
| Participant-wise variability | | | | | |
| SD of intercept | 0.39 | [0.35, 0.43] | 0.36 | [0.30, 0.44] | |
| SD of Δ-uncertainty | 0.92 | [0.81, 1.04] | 1.09 | [0.89, 1.35] | |
| SD of Δ-uncertainty × overall uncertainty | 12.85 | [0.56, 41.41] | 8.97 | [0.46, 28.35] | |
| SD of transformed threshold | 0.45 | [0.38, 0.54] | 0.35 | [0.26, 0.47] | |

This model can be summarized with the following formula: $logit(P(table\ on\ right\ chosen)) = Intercept + b_1 * \Delta\ uncertainty + b_2 * (overall\ uncertainty - \omega) * step(overall\ uncertainty - \omega) * \Delta\ uncertainty$, where step is the step function, $\omega = -2ln(0.5) * inv\_logit(\alpha)$. The intercept and parameters $b_1$, $b_2$, and α all vary by participant. This model was fit as a logistic regression.

**Appendix 3—table 7.** Test performance as a function of the tendency to approach uncertainty in exploration.

| Term | Pre-registered sample (194 participants) | | Preliminary sample (62 participants) | | |
|------|---------|---------|---------|---------|-------|
| | Median | 95% PI | Median | 95% PI | Units |
| Predictors | | | | | |
| Intercept | 1.56 | [1.51, 1.61] | 1.62 | [1.53, 1.72] | logit |
| Approach tendency | 2.96 | [2.67, 3.25] | 3.09 | [2.65, 3.57] | logit² / nat |

*Appendix 3—table 7 Continued on next page*

*Appendix 3—table 7 Continued*

| | Pre-registered sample | Preliminary sample |
|---|---|---|

The model can be summarized with the following R syntax formula: *test accuracy ~ 1 + approach tendency*. For the tendency to approach uncertainty, we computed the mean posterior approach parameter for each participant in the model described in **Appendix 3—table 6**. The model described here was fit as a logistic regression with binomial likelihood.

**Appendix 3—table 8.** Test performance as a function of tendency to avoid uncertainty in exploration when overall uncertainty is high.

| | Pre-registered sample | | Preliminary sample | | |
|---|---|---|---|---|---|
| | (194 participants) | | (62 participants) | | |
| Term | Median | 95% PI | Median | 95% PI | Units |
| Predictors | | | | | |
| Intercept | 1.52 | [1.48, 1.57] | 1.59 | [1.50, 1.68] | logit |
| Avoid tendency | 1.18 | [0.80, 1.58] | -0.52 | [–1.20, 0.21] | nat² |

The model can be summarized with the following R syntax formula: *test accuracy ~ 1 + approach tendency*. For the tendency to avoid uncertainty when overall uncertainty is high, we computed for each participant the area under the curve of the uncertainty approach/avoid graph, averaging across the posterior of the model described in **Appendix 3—table 6**. The model described here was fit as a logistic regression with binomial likelihood.

**Appendix 3—table 9.** Test performance as a function of tendency to approach uncertainty and the tendency to avoid uncertainty when overall uncertainty is high.

| | Pre-registered sample | | Preliminary sample | | |
|---|---|---|---|---|---|
| | (194 participants) | | (62 participants) | | |
| Term | Median | 95% PI | Median | 95% PI | Units |
| Predictors | | | | | |
| Intercept | 1.56 | [1.52, 1.61] | 1.62 | [1.53, 1.72] | logit |
| Approach tendency | 3.22 | [2.89, 3.55] | 3.10 | [2.64, 3.58] | logit² / nat |
| Avoid tendency | -0.83 | [-1.28,–0.40] | 0.02 | [–0.64, 0.75] | nat² |
| Participant-wise variability | | | | | |

The model can be summarized with the following R syntax formula:
*test > accuracy ~ 1 + avoid > tendency + approach > tendency*. For the tendency to avoid uncertainty when overall uncertainty is high, we computed for each participant the area under the curve of the uncertainty approach/avoid graph, averaging across the posterior of the model described in **Appendix 3—table 6**. The model described here was fit as a logistic regression with binomial likelihood.

**Appendix 3—table 10.** Drift diffusion model of exploration-phase choice and RTs.

| | Pre-registered sample | | Preliminary sample | | |
|---|---|---|---|---|---|
| | (113,746 trials, 194 participants) | | (31,205 trials, 62 participants) | | |
| Term | Median | 95% PI | Median | 95% PI | Units |
| Predictors | | | | | |
| B - bound height | 0.74 | [0.72, 0.76] | 0.74 | [0.71, 0.76] | |
| $\mu_0$ - drift rate offset | -0.01 | [–0.06, 0.03] | -0.01 | [–0.08, 0.06] | |
| $\kappa$ - dependence of drift rate on uncertainty | 0.69 | [0.58, 0.78] | 0.78 | [0.58, 0.99] | |
| $t_{ND}$ - *non-decision time* | 0.28 | [0.26, 0.31] | 0.27 | [0.25, 0.31] | |
| Participant-wise variability | | | | | |

*Appendix 3—table 10 Continued on next page*

*Appendix 3—table 10 Continued*

|  | Pre-registered sample | | Preliminary sample | |
|---|---|---|---|---|
| SD of B | 0.12 | [0.11, 0.14] | 0.10 | [0.08, 0.12] |
| SD of $\mu_0$ | 0.30 | [0.27, 0.33] | 0.25 | [0.21, 0.31] |
| SD of $\kappa$ | 0.66 | [0.58, 0.74] | 0.76 | [0.62, 0.94] |
| SD of $t_{ND}$ | 0.16 | [0.14, 0.19] | 0.12 | [0.10, 0.15] |

We used a drift-diffusion model to formalize the dependence of RTs and choice on evidence. A drift-diffusion model is one variant in the sequential sampling family of models. The model posits that samples of momentary evidence are integrated over time. The expectation of the momentary evidence distribution is termed the drift rate μ, and its standard deviation is termed the diffusion coefficient. The decision is made when integrated evidence reaches an upper or lower bound (±B), whose sign determines the choice. Processes external to decision making are modeled by $t_N D$, a constant added to the RT. In this model, μ is allowed to depend linearly on Δ-uncertainty, $\mu = \mu_0 + \kappa \cdot \Delta - uncertainty$, such that $\kappa$ captures the dependence of drift rate on Δ-uncertainty, and $\mu_0$ is a general bias to make rightward or leftward choices.

Prior to fitting the model to the data, we excluded trials for which overall uncertainty was above the threshold estimated for each participant by the model described in **Appendix 3—table 6**. As we find qualitatively different choice behavior above the threshold, we couldn't justify modeling these trials together with the majority of trials. Fitting a piecewise regression DDM model was beyond the capabilities of current software.

**Appendix 3—table 11.** Test performance as a function of drift diffusion model parameters for exploration phase.

|  | Pre-registered sample | | Preliminary sample | | |
|---|---|---|---|---|---|
|  | (113,746 trials, 194 participants) | | (31,205 trials, 62 participants) | | |
| Term | Median | 95% PI | Median | 95% PI | Units |
| Predictors | | | | | |
| Intercept | 0.33 | [–0.34, 1.01] | 0.23 | [–0.86, 1.28] | |
| Final uncertainty | -0.87 | [-0.98,–0.76] | -0.93 | [-1.13,–0.74] | |
| B - bound height | 1.46 | [0.58, 2.34] | 1.49 | [0.05, 2.90] | |
| $\kappa$ - dependence of drift rate on uncertainty | 0.81 | [0.58, 1.07] | 0.88 | [0.57, 1.21] | |
| Participant-wise variability | | | | | |
| SD of intercept | 0.87 | [0.74, 1.01] | 0.68 | [0.47, 0.96] | |
| SD of final uncertainty | 0.49 | [0.39, 0.60] | 0.44 | [0.24, 0.67] | |
| Correlation of intercept and final uncertainty | -0.81 | [-0.92,–0.66] | -0.83 | [-0.98,–0.44] | |

This model can be summarized with the following R syntax formula: *test accuracy ~ 1 + final uncertainty + B + κ + (1 + final uncertainty|participant)*. This model was fit as a logistic regression. As B and $\kappa$ are parameters estimated from the model described in **Appendix 3—table 10**, we took into account our error in measuring them when using them as predictors in this model. Thus, the posterior distribution for each participant's B and $\kappa$ parameters was summarized as a mean and standard deviation. These summary statistics were used to approximate the posterior as a normal distribution from which a latent variable was drawn during the estimation of this model. This method propagates the uncertainty in the values of B and $\kappa$ into the estimates reported here. Prior to using this method, we inspected the posteriors from the model summarized in **Appendix 3—table 10**, and made sure the normal distribution is an adequate approximation for these posteriors.

**Appendix 3—table 12.** Exploration-phase choices as a function of Δ-uncertainty, overall uncertainty, and side of repeat option.

| Term | Pre-registered sample (146,766 trials, 194 participants) | | Preliminary sample (41,009 trials, 62 participants) | | |
| --- | --- | --- | --- | --- | --- |
| | Median | 95% PI | Median | 95% PI | Units |
| **Predictors** | | | | | |
| Intercept | –0.04 | [–0.10, 0.02] | –0.03 | [–0.14, 0.07] | logit |
| Δ-uncertainty | 1.01 | [0.86, 1.14] | 1.16 | [0.87, 1.44] | logit / nat |
| Δ-uncertainty × overall uncertainty | –326.14 | [-468.61,–245.18] | –349.54 | [-691.47,–251.13] | logit / nat² |
| Transformed threshold | 2.55 | [2.40, 2.72] | 2.36 | [2.16, 2.67] | a.u. |
| Repeat choice on right | 0.50 | [0.42, 0.59] | 0.57 | [0.43, 0.72] | logit difference |
| Participant-wise variability | | | | | |
| SD of intercept | 0.40 | [0.36, 0.44] | 0.38 | [0.31, 0.46] | |
| SD of Δ-uncertainty | 0.90 | [0.80, 1.02] | 1.05 | [0.86, 1.31] | |
| SD of Δ-uncertainty × overall uncertainty | 14.21 | [0.98, 41.45] | 9.23 | [0.46, 29.28] | |
| SD of transformed threshold | 0.44 | [0.36, 0.54] | 0.35 | [0.22, 0.50] | |
| SD of repeat choice | 0.57 | [0.51, 0.64] | 0.53 | [0.44, 0.66] | |
| Correlation of intercept and Δ-uncertainty | –0.06 | [–0.21, 0.10] | 0.05 | [–0.21, 0.32] | |
| Correlation of intercept and Δ-uncertainty×overall uncertainty | –0.08 | [–0.76, 0.72] | –0.01 | [–0.76, 0.75] | |
| Correlation of Δ-uncertainty and Δ-uncertainty×overall uncertainty | 0.01 | [–0.69, 0.69] | –0.10 | [–0.78, 0.70] | |
| Correlation of intercept and repeat choice | –0.16 | [-0.30,–0.00] | –0.06 | [–0.31, 0.22] | |
| Correlation of Δ-uncertainty and repeat choice | 0.32 | [0.17, 0.46] | 0.11 | [–0.18, 0.38] | |
| Correlation of Δ-uncertainty × overall uncertainty and repeat choice | 0.38 | [–0.67, 0.87] | 0.00 | [–0.76, 0.77] | |
| Correlation of intercept and threshold | –0.03 | [–0.26, 0.22] | 0.31 | [–0.09, 0.64] | |
| Correlation of Δ-uncertainty and threshold | –0.35 | [-0.52,–0.16] | 0.09 | [–0.27, 0.43] | |
| Correlation of Δ-uncertainty × overall uncertainty and threshold | –0.42 | [–0.88, 0.61] | –0.10 | [–0.79, 0.70] | |
| Correlation of repeat choice and threshold | –0.60 | [-0.74,–0.43] | –0.38 | [-0.64,–0.05] | |

This model can be summarized with the following formula:
$logit(P(table\ on\ right\ chosen)) = Intercept + b1 \cdot \Delta-uncertainty + b2 \cdot (overall\ uncertainty - \omega)$
$\cdot step(overall\ uncertainty - \omega) \cdot \Delta-uncertainty + b3 \cdot repeat\ on\ right$, where step is the step function,
$\omega = -2ln(0.5) \cdot inv\_logit(\alpha)$. The intercept and parameters b1, b2, b3, and α all vary by participant, and their correlations across participants are modeled. This model was fit as a logistic regression.

**Appendix 3—table 13.** Comparing models predicting exploration-phase choice from the three hypothesized strategies and the tendency to repeat previous choices.

| Predictors included | Pre-registered sample ELPD difference (SE) | Preliminary sample ELPD difference (SE) |
|---|---|---|
| Uncertainty and tendency to repeat | 0 | 0 |
| EIG and tendency to repeat | −375.97 (49.66) | 91.03 (31.64) |
| Exposure and tendency to repeat | −1022.84 (61.47) | −443.51 (38.03) |
| Uncertainty | −2639.82 (71.53) | −780.11 (38.98) |
| EIG | −3090.33 (87.49) | −986.96 (50.48) |
| Exposure | −3123.77 (91.03) | −1061.01 (52.67) |

Comparing the three models described in **Appendix 3—tables 3–5**, and the same models with the addition of a term coding for whether the repeat choice was presented on the right. Values represent the difference in expected log predictive density (ELPD) from the best fitting model. For both samples, adding the choice repetition term improves model fit substantially, while not changing the order of the three strategies.

**Appendix 3—table 14.** Drift diffusion model of exploration-phase choice and RTs, differentiating between repeat and switch choices.

| | Pre-registered sample | | Preliminary sample | | |
|---|---|---|---|---|---|
| | (113,746 trials, 194 participants) | | (31,205 trials, 62 participants) | | |
| Term | Median | 95% PI | Median | 95% PI | Units |
| Predictors | | | | | |
| $B_0$- average bound height | 0.75 | [0.73, 0.76] | 0.74 | [0.72, 0.77] | |
| $B_{repeat}$- difference in bound height between repeat and switch chosen | -0.05 | [-0.05,–0.04] | -0.04 | [-0.06,–0.02] | |
| $\mu_0$- drift rate offset | -0.01 | [–0.06, 0.03] | -0.01 | [–0.08, 0.05] | |
| $\kappa$- dependence of drift rate on uncertainty | 0.70 | [0.61, 0.80] | 0.81 | [0.60, 1.01] | |
| $\kappa_{repeat}$- difference in dependence between repeat and switch chosen | -0.32 | [-0.43,–0.22] | -0.28 | [-0.49,–0.08] | |
| $t_{ND}$- non-decision time | 0.28 | [0.26, 0.31] | 0.28 | [0.25, 0.31] | |
| Participant-wise variability | | | | | |
| SD of $B_0$ | 0.12 | [0.11, 0.14] | 0.10 | [0.08, 0.12] | |
| SD of $B_{repeat}$ | 0.05 | [0.04, 0.05] | 0.05 | [0.04, 0.07] | |
| SD of $\mu_0$ | 0.30 | [0.27, 0.33] | 0.26 | [0.21, 0.31] | |
| SD of $\kappa$ | 0.64 | [0.57, 0.72] | 0.74 | [0.60, 0.92] | |
| SD of $\kappa_{repeat}$ | 0.50 | [0.40, 0.60] | 0.58 | [0.39, 0.80] | |
| SD of $t_{ND}$ | 0.16 | [0.14, 0.19] | 0.12 | [0.10, 0.15] | |

We refit the DDM described in **Appendix 3—table 10**, allowing both B and $\kappa$ to vary by whether the choice was a repeat or switch choice (see main text for definition).

**Appendix 3—table 15.** Test performance as a function of the tendency to repeat exploration-phase choices.

| | Pre-registered sample | | Preliminary sample | | |
|---|---|---|---|---|---|
| | (194 participants) | | (62 participants) | | |
| Term | Median | 95% PI | Median | 95% PI | Units |
| Predictors | | | | | |

*Appendix 3—table 15 Continued on next page*

*Appendix 3—table 15 Continued*

|  | Pre-registered sample | | Preliminary sample | | |
|---|---|---|---|---|---|
| Intercept | 1.52 | [1.47, 1.56] | 1.59 | [1.50, 1.67] | logit |
| Tendency to repeat | 0.09 | [0.07, 0.11] | 0.11 | [0.06, 0.16] | logit / logit |
| Participant-wise variability | | | | | |

The model can be summarized with the following R syntax formula: ***test accuracy ∼ 1 + tendency to repeat***. For the tendency to repeat, we computed the mean posterior parameter for each participant in the model described in ***Appendix 3—tables 1 and 2*** This model was fit as a logistic regression with binomial likelihood.

**Appendix 3—table 16.** Exploration-phase RTs as a function of memory lag and side of repeat option.

|  | Pre-registered sample | | Preliminary sample | | |
|---|---|---|---|---|---|
|  | (126,848 trials, 194 participants) | | (35,264 trials, 62 participants) | | |
| Term | Median | 95% PI | Median | 95% PI | Units |
| Predictors | | | | | |
| Intercept | -0.34 | [-0.38,–0.30] | -0.35 | [-0.42,–0.30] | log s |
| Memory lag | 0.02 | [0.02, 0.03] | 0.03 | [0.02, 0.03] | log s / trial |
| Repeat choice on right | -0.05 | [-0.06,–0.04] | -0.05 | [-0.07,–0.03] | log s difference |
| Memory lag × repeat | 0.02 | [0.02, 0.03] | 0.02 | [0.02, 0.03] | 1/trial |
| Participant-wise variability | | | | | |
| SD of intercept | 0.29 | [0.26, 0.31] | 0.23 | [0.19, 0.27] | |
| SD of memory lag | 0.02 | [0.01, 0.02] | 0.02 | [0.01, 0.02] | |
| SD of repeat choice on right | 0.06 | [0.05, 0.07] | 0.07 | [0.06, 0.09] | |
| SD of memory lag ×repeat | 0.02 | [0.01, 0.02] | 0.02 | [0.01, 0.03] | |
| Correlation of intercept and memory lag | 0.41 | [0.24, 0.56] | 0.19 | [–0.11, 0.46] | |
| Correlation of intercept and repeat | -0.27 | [-0.42,–0.11] | -0.35 | [-0.57,–0.07] | |
| Correlation of memory lag and repeat | -0.70 | [-0.83,–0.53] | -0.27 | [–0.57, 0.07] | |
| Correlation of intercept and memory lag × repeat | 0.27 | [0.04, 0.48] | 0.26 | [–0.17, 0.65] | |
| Correlation of memory lag and memory lag × repeat | 0.75 | [0.51, 0.90] | 0.21 | [–0.28, 0.65] | |
| Correlation of repeat and memory lag × repeat | -0.91 | [-0.98,–0.77] | -0.74 | [-0.94,–0.36] | |

The model can be summarized with the following R syntax formula: ***log RT ∼ 1 + memory lag ∗ repeat on right +***

***(1 + memory lag · repeat on right|participant)***. This model was fit as a lognormal regression.

**Appendix 3—table 17.** Exploration-phase choices as a function of Δ-uncertainty, memory lag, and side of repeat option.

|  | Pre-registered sample | | Preliminary sample | | |
|---|---|---|---|---|---|
|  | (126,973 trials, 194 participants) | | (35,304 trials, 62 participants) | | |
| Term | Median | 95% PI | Median | 95% PI | Units |
| Predictors | | | | | |

*Appendix 3—table 17 Continued on next page*

*Appendix 3—table 17 Continued*

| | Pre-registered sample | | Preliminary sample | | |
|---|---|---|---|---|---|
| Intercept | -0.03 | [–0.09, 0.02] | -0.02 | [–0.12, 0.08] | logit |
| Δ-uncertainty | 1.03 | [0.89, 1.17] | 1.16 | [0.87, 1.43] | logit / nat |
| Memory lag | -0.01 | [–0.02, 0.00] | 0.01 | [–0.00, 0.02] | logit / trial |
| Repeat choice on right | 0.45 | [0.37, 0.52] | 0.50 | [0.37, 0.63] | logit difference |
| Δ-uncertainty × memory lag | -0.08 | [-0.11,–0.04] | -0.14 | [-0.20,–0.07] | logit / nat * trial |
| Memory lag × repeat | -0.13 | [-0.15,–0.11] | -0.08 | [-0.12,–0.04] | 1/trial |
| Participant-wise variability | | | | | |
| SD of intercept | 0.40 | [0.36, 0.45] | 0.37 | [0.31, 0.45] | |
| SD of Δ-uncertainty | 0.92 | [0.81, 1.04] | 1.06 | [0.87, 1.32] | |
| SD of memory lag | 0.03 | [0.02, 0.04] | 0.02 | [0.00, 0.04] | |
| SD of repeat choice on right | 0.49 | [0.44, 0.55] | 0.45 | [0.36, 0.57] | |
| SD of Δ-uncertainty×memory lag | 0.13 | [0.08, 0.17] | 0.07 | [0.00, 0.19] | |
| SD of memory lag × repeat | 0.10 | [0.08, 0.13] | 0.11 | [0.07, 0.16] | |

The model can be summarized with the following R syntax formula: *table on right chosen ∼ 1 + Δ − uncertainty · memory lag + memory lag : repeat on right + (1 + Δ − uncertainty · memory lag + memory lag : repeat on right|participant)*. This model was fit as a logistic regression. For brevity, the correlations in participant-wise variability are omitted from this table.

**Appendix 3—table 18.** Exploration-phase choices as a function of Δ-uncertainty, overall uncertainty, and trial number.

| | Pre-registered sample | | Preliminary sample | | |
|---|---|---|---|---|---|
| | (146,766 trials, 194 participants) | | (41,009 trials, 62 participants) | | |
| Term | Median | 95% PI | Median | 95% PI | Units |
| Predictors | | | | | |
| Intercept | -0.04 | [–0.09, 0.02] | -0.03 | [–0.12, 0.06] | logit |
| Δ-uncertainty | 1.00 | [0.85, 1.14] | 1.16 | [0.85, 1.46] | logit / nat |
| Δ-uncertainty × overall uncertainty | -480.86 | [-628.28,–379.29] | -450.20 | [-568.82,–356.27] | logit / nat² |
| Transformed threshold | 2.56 | [2.43, 2.69] | 2.32 | [2.18, 2.48] | a.u. |
| Δ-uncertainty × trial # | -0.00 | [–0.00, 0.00] | -0.00 | [–0.01, 0.00] | logit / nat ×trial |
| Participant-wise variability | | | | | |
| SD of intercept | 0.39 | [0.35, 0.43] | 0.36 | [0.30, 0.45] | |
| SD of Δ-uncertainty | 0.94 | [0.83, 1.06] | 1.13 | [0.93, 1.39] | |
| SD of Δ-uncertainty × overall uncertainty | 10.68 | [0.52, 36.54] | 8.87 | [0.45, 28.51] | |
| SD of transformed threshold | 0.43 | [0.36, 0.51] | 0.34 | [0.25, 0.46] | |
| SD Δ-uncertainty × trial # | 0.02 | [0.01, 0.02] | 0.02 | [0.01, 0.02] | |

*Appendix 3—table 18 Continued on next page*

*Appendix 3—table 18 Continued*

|  | Pre-registered sample | Preliminary sample |
|---|---|---|

We refit the piecewise regression model described in **Appendix 3—table 6**, accounting for a possible interaction between Δ-uncertainty and trial number. We find no significant interaction in the pre-registered sample, nor the preliminary sample. All other terms in the model remained practically the same.

The model can be summarized with the following formula: $logit(P(table\ on\ right\ chosen)) = Intercept + b1 \cdot \Delta-uncertainty + b2 \cdot (overall\ uncertainty - \omega) \cdot step(overall\ uncertainty - \omega) \cdot \Delta-uncertainty + b4 \cdot trial\#\cdot\Delta-uncertainty$, where step is the step function, $\omega = -2ln(0.5) \cdot inv\_logit(\alpha)$. The intercept and parameters b1, b2, b4, and α all vary by participant. This model was fit as a logistic regression.

**Appendix 3—table 19.** Model comparison for sequential sampling models of the tendency to repeat previous choices.

| Parameters varying by repeat / switch choice | Pre-registered sample DIC | Preliminary sample DIC |
|---|---|---|
| None | 218,877.63 | 59,365.40 |
| $\kappa$ - the dependence of RT on Δ-uncertainty | 218,666.99 | 59,311.25 |
| $\kappa$ - the dependence of RT on Δ-uncertainty | 218,666.99 | 59,311.25 |
| B - Bound height | 217,928.89 | 59,096.38 |
| Both $\kappa$ and B | 217,669.43 | 59,046.67 |

The model reported in **Appendix 3—table 14** captures the tendency to repeat previous choices by allowing both the dependence of RT on Δ-uncertainty and the bound height parameters to vary by whether the choice was a repeat or switch choice (last row in this table). Here, it is compared against the simpler models nested within it. For both samples, the full model is favored over the partial models, as is indicated by lower deviance information criterion (DIC) values. DIC values are derived from the likelihood of the data given estimated parameters, and the effective number of parameters in the model. Absolute values of DIC depend on sample size and the attributes of the noise distribution. Accordingly, DIC values should only be compared between models fit to the same dataset.

**Appendix 3—table 20.** Test performance as a function of exploration-phase uncertainty coefficient.

|  | Pre-registered sample | | Preliminary sample | | |
|---|---|---|---|---|---|
|  | (194 participants) | | (62 participants) | | |
| Term | Median | 95% PI | Median | 95% PI | Units |
| Predictors |  |  |  |  |  |
| Intercept | 1.56 | [1.51, 1.61] | 1.61 | [1.52, 1.70] | logit |
| Coefficient for uncertainty | 2.90 | [2.62, 3.19] | 2.72 | [2.29, 3.15] | logit² / nat |

The model can be summarized with the following R syntax formula: **test accuracy ∼ 1 + uncertainty** , where uncertainty is the mean posterior parameter for each participant in the model described in **Appendix 3—table 3**. The model described here was fit as a logistic regression with binomial likelihood.

**Appendix 3—table 21.** Test performance as a function of exploration-phase EIG coefficient.

|  | Pre-registered sample | | Preliminary sample | | |
|---|---|---|---|---|---|
|  | (194 participants) | | (62 participants) | | |
| Term | Median | 95% PI | Median | 95% PI | Units |
| Predictors |  |  |  |  |  |
| Intercept | 1.57 | [1.52, 1.62] | 1.61 | [1.52, 1.71] | logit |
| Coefficient for EIG | 1.98 | [1.77, 2.22] | 1.72 | [1.37, 2.11] | logit² / nat |

The model can be summarized with the following R syntax formula: **test accuracy ∼ 1 + EIG**, where EIG is the mean posterior parameter for each participant in the model described in **Appendix 3—table 4**. The model described here was fit as a logistic regression with binomial likelihood.

**Appendix 3—table 22.** Test performance as a function of exploration-phase exposure coefficient.

| | Pre-registered sample (194 participants) | | Preliminary sample (62 participants) | | |
|---|---|---|---|---|---|
| Term | Median | 95% PI | Median | 95% PI | Units |
| Predictors | | | | | |
| Intercept | 1.52 | [1.48, 1.57] | 1.58 | [1.50, 1.68] | logit |
| Coefficient for exposure | -0.18 | [–0.43, 0.06] | -1.09 | [-1.56,–0.61] | logit² / trial |

The model can be summarized with the following R syntax formula: ***test accuracy*** $\sim$ ***1 + exposure***, where exposure is the mean posterior parameter for each participant in the model described in ***Appendix 3—table 5***. The model described here was fit as a logistic regression with binomial likelihood.

**Appendix 3—table 23.** Model comparison for models predicting test accuracy from coefficients for each of the three strategies.

| Coefficient predicting test accuracy | Pre-registered sample ELPD difference (SE) | Preliminary sample ELPD difference (SE) |
|---|---|---|
| Uncertainty | 0 | 0 |
| EIG | −37.53 (31.78) | −31.96 (19.08) |
| Exposure | −232.81 (45.17) | −86.46 (24.14) |

Comparing the three models described in ***Appendix 3—tables 20–22***. Values represent the difference in expected log predictive density (ELPD) from the best fitting model. For both samples, individual differences in the uncertainty coefficient bet predict test accuracy.

