## [Editor Report · eLife Assessment]

This study presents an **important** investigation of how people approach and avoid uncertainty, with a particular focus on the effects of overall uncertainty. They find that individuals approach uncertainty to a point, but when uncertainty is particularly high, they avoid it. The results are interpreted under a cognitive cost-resource rational framework. The methods are **convincing**, using appropriate and current methodologies.

---

## [Referee Report · Reviewer #1 (Public review)]

This manuscript reports on the behavior of participants playing a game to measure exploration. Specifically, participants completed a task with blocks of exploratory choices (choosing between two 'tables', and within each table, two 'card decks', each of which had a specific probability of showing cards with one color versus another) and test choices, where participants were asked to choose which of the two decks per table had a higher likelihood of one color. Blocks differed on how long (how many trials) the exploration phase lasted. Participants' choices were fit to increasingly complex models of next-trial exploration. Participants' choices were best fit by an intermediate model where the difference in uncertainty between tables influenced the choice. Next, the authors investigated factors affecting whether participants sought out or avoided uncertainty, their choice reaction times, and the relationship of these measures with performance during the test phase of each block. Participants were uncertainty-seeking (exploratory) under most levels of overall uncertainty but became less uncertainty-seeking at high levels of total uncertainty. Participants with a stronger tendency to approach uncertainty at lower levels of total uncertainty were more accurate in the test phase, while the tendency to avoid uncertainty when total uncertainty was high was also weakly positively related to test accuracy. In terms of reaction times, participants whose reaction times were more related to the level of uncertainty, and who deliberated longer, performed better. The individual tendency to repeat choices was related to avoidance of uncertainty under high total uncertainty and better test performance. Lastly, choices made after a longer lag were less affected by these measures.

---

## [Author Response]

The following is the authors’ response to the original reviews

We would like to sincerely thank the editor and reviewers for their thoughtful and constructive feedback on our manuscript. We are grateful not only for the close reading and insightful suggestions, but also for the open and generous way in which the reviewers engaged with our work.In revising the manuscript, we have clarified how our contribution is situated within the existing literature, conducted additional analyses to examine individual differences in exploration strategies, expanded and refined our description of the DDM analyses, and added correlations between strategies and other behavioral measures. We have also clarified methodological points, such as the estimation of thresholds, and provided new supplementary figures and analyses where appropriate. In several places, we have modified and qualified our interpretations in line with the reviewers’ comments.We believe these changes have significantly strengthened the manuscript, and we are grateful for the scientific dialogue with the reviewers.

**Review 1 (Public review):**
This manuscript reports on the behavior of participants playing a game to measure exploration. Specifically, participants completed a task with blocks of exploratory choices (choosing between two 'tables', and within each table, two 'card decks', each of which had a specific probability of showing cards with one color versus another) and test choices, where participants were asked to choose which of the two decks per table had a higher likelihood of one color. Blocks differed on how long (how many trials) the exploration phase lasted. Participants' choices were fit to increasingly complex models of next-trial exploration. Participants' choices were best fit by an intermediate model where the difference in uncertainty between tables influenced the choice. Next, the authors investigated factors affecting whether participants sought out or avoided uncertainty, their choice reaction times, and the relationship of these measures with performance during the test phase of each block. Participants were uncertainty-seeking (exploratory) under most levels of overall uncertainty but became less uncertainty-seeking at high levels of total uncertainty. Participants with a stronger tendency to approach uncertainty at lower levels of total uncertainty were more accurate in the test phase, while the tendency to avoid uncertainty when total uncertainty was high was also weakly positively related to test accuracy. In terms of reaction times, participants whose reaction times were more related to the level of uncertainty, and who deliberated longer, performed better. The individual tendency to repeat choices was related to avoidance of uncertainty under high total uncertainty and better test performance. Lastly, choices made after a longer lag were less affected by these measures.The authors note that their paradigm, which does not provide immediate rewarding feedback, is novel. However, the resulting behavior appears similar to other exploratory learning tasks, so it's unclear what this task design adds - besides perhaps showing that exploratory behavior is similar across types of reward environments. Several papers have shown that cognitive constraints modulate exploration (PMIDs: 30667262, 24664860, 35917612, 35260717); although this paper provides novel insights, it does not situate its findings in the context of this prior literature. As a result, what it adds to the literature is difficult to discern.

We are grateful for your thoughtful reading of our paper and for pointing us to these relevant references. We appreciate the need to clarify how our work is situated within the existing literature. In brief, the novelty of our paper lies in measuring exploration in contexts where it is not in direct competition with the need to exploit knowledge for reward. This approach enables us to include orders of magnitude more exploration trials. With this increased power, we were able— for the first time— to distinguish between competing algorithms for addressing uncertainty, and we identified a novel tendency to avoid uncertainty when overall uncertainty is high. We now state this more clearly in the discussion section and cite the suggested papers.

“While the literature on exploration is expansive, the paradigm presented here extends it in important ways. Researchers of reinforcement learning have previously examined exploration in the context of reward-seeking decisions. Using such paradigms as the bandit task Schulz and Gershman (2019), it was demonstrated that humans don't always choose the option they believe will yield the most reward, but also make random and directed choices with the aim of exploring other uncertain options (Schulz and Gershman, 2019; Wilson et al., 2014). Recently, studies using the bandit task have lent empirical support to the notion that exploration is difficult, as participants explore less under time pressure or cognitive load (Brown et al., 2022; Otto et al., 2014; Cogliati Dezza et al., 2019; Wu et al., 2022). Crucially, this literature has focused on cases where reward can be gained on each trial (Brown et al., 2022; Cohen et al., 2007; Daw et al., 2006; Schulz and Gershman, 2019; Song et al., 2019; Tversky and Edwards, 1966; Wilson et al., 2014; Wu et al., 2022). In such tasks, the motivation to exploit current knowledge predominates exploration, rendering it rare and difficult to measure (Findling et al., 2019). In contrast, our task was designed to remove the impetus to immediately exploit current knowledge , and as a result we were able to observe many exploratory choices. With this increased experimental power, we were able to compare different algorithms approximating the goal of approaching uncertainty, and describe how and when humans avoid uncertainty instead of approaching it.”

**Reviewer #1 (Recommendations For The Authors):**
Are all participants best fit by the delta uncertainty model? Since other parts of the paper focus on individual differences, it would be useful to examine if people differ in the computational complexity of their exploration strategies and if this difference relates to other behavior.

We thank you for this helpful suggestion, which prompted us to conduct additional analyses. To address your question, we summarized point-wise predictive accuracy for each participant and compared it across the three models. The results are presented in the new Supplements 2 and 3 to Figure 6.

These analyses show that, for the vast majority of participants, uncertainty was favored over exposure as a choice strategy, and for a sizable majority, it was also favored over EIG. As detailed in Figure 6 and its supplements, 125 participants were best described by uncertainty relative to EIG, 58 by EIG, and 11 showed inconclusive results. Similarly, 96 participants were better fit by uncertainty than exposure, while an additional 72 had negative exposure coefficients (consistent with uncertainty-based choice). Exposure was supported for 26 participants.

We also examined how these strategies relate to other behavioral measures. Exposure was not strongly linked to test performance. EIG, by contrast, showed a positive association with test performance, perhaps because it is more closely correlated with uncertainty. Importantly, however, across posterior predictive checks in the main text and supplements, approaching uncertainty continues to provide the best overall description of participants’ strategies.

The authors construct a hierarchy of exploratory strategies. Perseveration/switching is also an explore/exploit strategy that would lie above random exploration in the authors' hierarchy.

We chose not to place perseveration within the hierarchy, as from a normative perspective it is not, strictly speaking, an exploration strategy. At its extreme, perseveration would lead a participant to repeatedly sample only one option, leaving the others entirely unexplored. Switching is represented in the hierachy by the equating exposure strategy – they are very similar.

For the analyses examining uncertainty seeking vs. aversion by total uncertainty, how was the cut point determined? Did this differ across people?

Thank you for highlighting the need for greater clarity on this point. The threshold was indeed fitted to the data and varied significantly across participants (see Table 6 in Appendix 3). For each participant, the threshold marks the point at which behavior shifts from approaching to avoiding uncertainty. This threshold is a key factor underlying individual differences in the tendency to avoid uncertainty when overall uncertainty is high, as illustrated in the analyses of Figure 6 and related results. We now make this point clearer in the methods section:

“To quantify how the influence of Δ-uncertainty on choice varied with overall uncertainty, we fit a multilevel piecewise logistic regression model. This model estimated a threshold in overall uncertainty, treated as a free parameter, and allowed the slope of Δ-uncertainty on choice to differ below and above this threshold. Below the threshold, a positive slope reflects a tendency to approach uncertainty; above the threshold, a negative interaction captures the tendency to avoid Δ-uncertainty with higher values of overall uncertainty.”

More details on the DDM analyses are needed - it's not clear how the outputs of the DDM correspond to what is stated in the text in the results.

We agree that the section detailing the DDM analyses could be clarified. We analyzed two key parameters of the DDM: the drift rate, which we interpret as reflecting the efficacy of deliberation over uncertainty, and the bound separation, which corresponds to the tendency to deliberate rather than respond quickly. Our results show that good learners exhibit both higher drift rates and higher bounds. When participants repeat a previous choice, both the drift rate and bounds are lower. We changed the way we report the results:

“We found that RTs indeed varied in relation to the absolute value of Δ-uncertainty as expected b=0.69, 95% PI=[0.58,0.78]. Crucially, a stronger dependence of RT on the absolute value of Δ-uncertainty predicted better performance at test (drift-rate and test performance association b=0.81, 95% PI=[0.58,1.07]). We further found that participants who tended to deliberate longer for the sake of accuracy also tended to perform better at test (bound height and test perfromance association b=1.46, 95% PI=[0.58,2.34]; Figure8c). In summary, participants who were better at deliberating about uncertainty during exploration, and who deliberated for longer, performed better at test. Thus, making good exploratory choices that lead to efficient learning involves prolonged deliberation.”

We also provide a detailed explanation of this correspondence in the Methods section:

“The DDM explains RTs as the culmination of three interpretable terms. The first is the efficacy of a participant’s thought process in furnishing relevant evidence for the decision - in our case the efficacy of choosing according to Δ-uncertainty (the drift rate in DDM parlance). The second term governs the participant’s speed-accuracy tradeoff by determining how much evidence they require to commit to a decision. This can also be thought of as how long a participant is willing to deliberate when a decision is difficult (bound height). Finally, the portion of the RT not linked to the deliberation process is captured by a third term (non-decision time).”

The authors note that "the three choice strategies prescribe different table choices on most trials" but (from what I can see) only provide a representative participant's plot in Figure 2. What was the overall correlation of predicted choices from the three models?

Thank you for pointing out this oversight. The correlations are now shown in the supplement to Figure 2. In brief, correlations between exposure and the other two strategies are low, while the correlation between EIG and uncertainty is moderate. These dependencies motivated our decision to fit a separate logistic regression model for each strategy and to compare strategies using formal model comparison and posterior predictive checks, rather than including them all in a single regression model.

It appears that the models are all constructed to predict table choices and not card deck choices. Can the authors clarify this? If so, what role do the card deck choices have?

Indeed, the manuscript focuses on table choices, as these are the choices of primary interest from an exploration perspective. It is most straightforward to define the three exploration strategies with respect to table choices, whereas for deck choices it is not clear how to define EIG in respect to the perforamnce at test. The hierarchical structure of the task was originally chosen to increase complexity, with the goal of creating a rich task that engages cognitive resources. We have not formally tested this assumption, and do not expect that the patterns we observe should be absent in a flat version of the task.

**Reviewer 2 (Public review):**
Summary:This paper focuses on an interesting question that has puzzled psychologists for decades, that is, why do people demonstrate a mix of uncertainty approach and avoidance behavior, given the fact that reducing uncertainty could always gain information and seems beneficial? This paper designed a novel task to demonstrate behavioral signatures of uncertainty approaching and avoidance during the exploration phase within the same task at both a within-subject and betweensubject level. On the algorithmic level, this paper compared four different implementations of uncertainty-guided exploration and found that the model sensitive to relative uncertainty provides the best fit for human behavior compared to its counterparts using expected information gain or past exposure. This paper then links people's uncertainty attitude with accuracy and finds that uncertainty avoidance during exploration does not impair task performance, implying that uncertainty avoidance may be the output of a resource-rational decision-making process. To examine this account, this paper uses reaction time as an independent proxy of costly deliberation and shows that people deliberate shorter when engaging in repetitive choice, which presumably saves cognitive resources. Finally, the paper shows that people's tendency to engage in repetitive choice correlates with their tendency to avoid uncertainty, which supports the argument that avoiding uncertainty could be a strategy developed under the constraint of limited cognitive resources.Strengths:One of the highlights of this paper, as mentioned in the previous paragraph, is that the authors can establish the existence of the uncertainty approach and avoidance behavior within the same task whereas previous work usually focuses on one of them. This dissociation allows the authors to examine what situational factor is related to the emergence of the act of avoiding uncertainty, and extract parameters describing participants' attitude towards uncertainty during baseline as well as during situations where uncertainty avoidance is more common. Besides documenting the existence of uncertainty avoidance behavior, this paper also tried to explain this behavior by proposing under the resource rational framework and has carefully quantified different aspects (e.g., accuracy; choice speed) of participants' behavior as well as examined their relationships. Though more experiments are needed to fully understand human uncertainty avoidance behavior, this paper has provided both empirical and theoretical contributions toward a mechanistic understanding of how people balance approaching and avoiding uncertainty.Weaknesses:I have a couple of concerns related to this paper. First, there seems to exist an anticorrelation between total uncertainty and absolute relative uncertainty (Figure 5 panel C, \delta uncertainty is restricted to a small range when total uncertainty is high). It seems to be a natural product of the exploration process since the high total uncertainty phase is usually the period where the participant knows little about either option, leading to a less distinguishable relative uncertainty. However, it remains unknown whether the documented uncertainty avoidance still applies when extrapolating to larger absolute relative uncertainty.

We sincerely thank you for your close reading of our manuscript and for highlighting its strengths. In the paradigm we study, overall and relative uncertainty are not anticorrelated. While the two are related—as in any finite-information exploration task, where the value of overall uncertainty constrains the possible range of relative uncertainty—they are not correlated and can therefore be used as predictors in a single regression model. We agree that strategies could differ substantially in a (near) infinite-information setting, such as when people seek semantic knowledge. The advantage of a finite-information task is its tractability, which enables the computational analyses we conducted. That said, the inherently greater intractability of an infinite-information task would likely alter human strategies, as it poses challenges both to participants and to researchers.

It would be great if the experiment allows for a manipulation of uncertainty in the middle of the experiment (e.g., introducing a new deck/informing that one deck has been updated)

We agree, and look forward to probing this question in the future. We’ve added the point to our discussion section:

“Our theoretical analysis and experiments leave several open questions. One concerns the relationship between overall uncertainty and time on task: in our paradigm, overall uncertainty was correlated with the number of cards observed. Although our findings remain robust when trial number is included as a covariate in the regression models, future work could more directly disentangle these factors by orthogonalizing overall uncertainty and elapsed time. This might be achieved, for instance, by manipulating overall uncertainty within a game—such as by introducing new tables or altering outcome probabilities mid-round.”

Relatedly, the current 'threshold' of uncertainty avoidance behavior, if I understand correctly, is found by empirically fitting participants' data. This brings the question: can we predict when people will demonstrate uncertainty avoidance behavior before collecting any data? Or, is it possible that by measuring some metrics related to cognitive cost sensitivity, we could predict the proportion of choices that participants will show uncertainty-avoidant behavior?

Thank you again for probing our thinking further. The threshold of uncertainty is indeed fitted on an individual basis using a hierarchical model. We believe there should be ways to predict it. In the current data, we find that it is correlated with the baseline tendency to approach uncertainty: in other words, participants who perform better show a slightly stronger tendency to avoid uncertainty when overall uncertainty is high. This underscores the complexity of identifying correlates of a coping strategy, as it is intricately linked to the difficulty being coped with. We speculate that working memory capacity may play an important role in this strategy, as well as the interplay between working memory–based learning and slower incremental learning mechanisms. Beyond speculation, however, we currently have no data to test these ideas.

Finally, regarding the analysis of different behavior patterns in the game, it seems that the authors try to link repetitive behavior, uncertainty attitude, and accuracy together by testing the correlation between the two of them. I wonder whether other multivariate statistical methods e.g., mediation analysis, will be better suited for this purpose.

This was a very insightful comment. We revisited the data and fitted test performance using a multiple regression model, predicting performance from the three exploration-phase strategies simultaneously: baseline tendency to approach uncertainty, tendency to avoid uncertainty when overall uncertainty is high, and tendency to repeat previous choices. When adjusting for the baseline tendency to approach, we find that the tendency to avoid uncertainty is indeed associated with a slight decrement in test performance. However, in our sample, the better learners—who are more effective at approaching uncertainty—also tend to avoid it when overall uncertainty is high. This nuance highlights the point discussed earlier. We find similar results when fitting the data with a mediation model, but we favour the multiple regression approach, since have no strong convictions about which exploration strategy causes another. We have detailed this analysis in the main text and have accordingly modified and qualified our interpretation of this finding:

“In contrast, the relationship between the tendency to avoid uncertainty and test performance was more nuanced. In both samples, participants who were more inclined to approach uncertainty also tended to avoid it when overall uncertainty was high r=0.43, p=5.42 x 10^-10^. Accordingly, avoidance was positively correlated with test performance at the population level b=1.18, 95% PI=[0.80, 1.58] Figure 7b; see Methods for parameter estimation. However, once we adjusted for the tendency to approach, avoidance was reliably associated with worse test performance b=-0.83, 95% PI=[-1.28,-0.40].”

**Reviewer #2 (Recommendations For The Authors):**
Could the authors elaborate more on why the negative relationship between exposure and choice (Figure 4a) is a natural phenomenon under the relative uncertainty model?

Indeed, we believe this is a natural phenomenon under the uncertainty model. When simulating an uncertainty-driven agent, the negative relationship arises naturally. We interpret this as the agent repeatedly pursuing tables that are more difficult to learn—those with smaller probability differences. The agent is drawn to these tables precisely because they are harder to master. By contrast, an EIG-driven agent would not repeatedly return to tables that are too difficult to learn. We have revised the Results section to make this point clearer:

“The simulations demonstrate that the surprising negative correlation between choice and Δ-exposure is an epiphenomenon of uncertainty-driven exploration: agents repeatedly return to harder-to-learn tables, gaining more exposure to them precisely because they remain more uncertain about these tables.”

It would be great if the authors could provide the correlation between different uncertainty estimates to help the readers have a better sense of how different these estimates are.

We’ve added this information in the supplement to Figure 2. In brief, correlations between exposure and the other two strategies are low, while the correlation between EIG and uncertainty is moderate. These dependencies motivated our decision to fit a separate logistic regression model for each strategy and to compare strategies using formal model comparison and posterior predictive checks, rather than including them all in a single regression model.